# BOOSTING LLM TRANSLATION SKILLS WITHOUT GENERAL ABILITY LOSS VIA RATIONALE DISTILLATION

## ABSTRACT

Large Language Models (LLMs) have achieved impressive results across numerous NLP tasks but still encounter difficulties in machine translation. Traditional methods to improve translation have typically involved fine-tuning LLMs using parallel corpora. However, vanilla fine-tuning often leads to catastrophic forgetting of the instruction-following capabilities and alignment with human preferences, compromising their broad general abilities and introducing potential security risks. These abilities, which are developed using proprietary and unavailable training data, make existing continual instruction tuning methods ineffective. To overcome this issue, we propose a novel approach called **RaDis** (**Ra**tionale **Dis**tillation). RaDis harnesses the strong generative capabilities of LLMs to create rationales for training data, which are then "replayed" to prevent forgetting. These rationales *connect prior knowledge with new tasks*, acting as *self-distillation targets* to regulate the training process. By jointly training on reference translations and self-generated rationales, the model can learn new translation skills while preserving its general abilities. Extensive experiments demonstrate that our method enhances machine translation performance while maintaining the broader capabilities of LLMs across other tasks. This work presents a pathway for creating more versatile LLMs that excel in specialized tasks without compromising generality or safety and *provides a fresh angle for utilizing rationales in the CL field.*

## 1 INTRODUCTION

Large Language Models (LLMs) have demonstrated exceptional performance across diverse Natural Language Processing (NLP) tasks. However, in the realm of Machine Translation (MT), they still fall short compared to conventional supervised encoder-decoder models (Xu et al., 2024a). Recent studies have sought to enhance the translation performance of LLMs through continual instruction-tuning with parallel corpora (Yang et al., 2023; Xu et al., 2024a). While this approach effectively boosts translation performance, it often comes at the cost of the inherent general ability and safety alignment of LLMs. As illustrated in Figure 1, fine-tuning LLaMA-2-Chat and Mistral-v0.2-Instruct results in a significant decline in these models' performance on MT-Bench (Zheng et al., 2023). This phenomenon is known as Catastrophic Forgetting (CF) (French, 1993), which remains the major obstacle to developing models that seamlessly integrate strong translation performance with broader general-purpose utility.

Various Continual Learning (CL) approaches have been proposed to mitigate CF (Chen & Liu, 2018). In the context of LLMs, replay-based methods (Scialom et al., 2022; Yin et al., 2022; Mok et al., 2023; He et al., 2024), which store small subsets of previous data for rehearsal, are often favored for their simplicity and effectiveness. However, a critical limitation of replay-based methods is their reliance on access to the original training data, which is frequently unavailable in real-world applications. This limitation greatly reduces their feasibility in scenarios like the one in this paper, where the goal is to boost translation skills while preserving the general abilities of open-sourced LLMs, which are gained from proprietary, in-house data. Some studies have incorporated open-source general instruction-following data as a substitute (Jiao et al., 2023; Zhang et al., 2023). However, the limited quality of these open-source datasets results in performance that significantly falls short of instruction-tuned LLMs.

To address this problem, this paper explores leveraging the strong generative ability of LLMs to *synthesize their own replay data*. However, given the vast task space of LLMs and the limited data we could use, generating high-quality synthesis data that encapsulates diverse general knowledge remains a non-trivial question. We draw inspiration from an observation that instruction-tuned LLMs are capable of generating detailed **rationales** when tasked with translation requests (see Section 3.1 and Appendix E for more details). Previous studies in the reasoning field suggest that rationales generated by LLMs contain valuable knowledge and can be used to distill reasoning abilities (Wadhwa et al., 2024; Xu et al., 2024b). In line with these findings, we show that self-generated rationales *encapsulate the internal general knowledge* leveraged during translation that *connect prior knowledge with new tasks*, acting as *self-distillation targets* to alleviate forgetting.

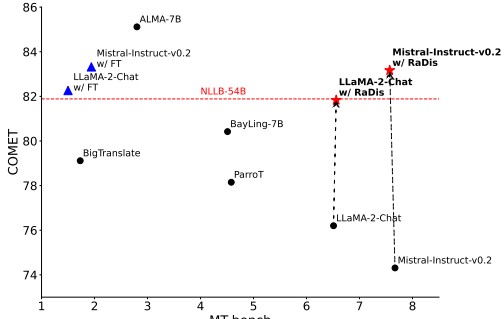

Figure 1: Translation performance (COMET) and general conversational and instruction-following ability (MT-Bench). While both Fine-tuning (blue triangle) and RaDis (red star) greatly enhance the translation performance, RaDis helps preserve most of the models' general ability.

Thus, we propose a novel training method named **RaDis** (**Ra**tionale **Dis**tillation). It prompts the LLM to generate rationales for the reference translations in original training data, and then concatenates references and rationales, forming an enriched dataset for subsequent training. By incorporating both the rationales and the references into the training, RaDis builds an underlying reasoning framework that ties previous knowledge with new tasks and introduces a self-distillation loss on the rationale, thereby mitigating the forgetting issue.

Comprehensive experiments using two widely adopted LLMs, LLaMA-2-7B-Chat (Touvron et al., 2023) and Mistral-7B-Instruct-v0.2 (Jiang et al., 2023) validates the effectiveness of RaDis. As depicted in Figure 1, RaDis enhances translation performance by 5.6 and 8.9 COMET points, which is comparable to vanilla fine-tuning, while preserving the models' original performance on general ability benchmarks. Further analysis reveals that distilling self-generated rationales not only outperforms distilling from external rationales generated by a much stronger model but also avoids the conflict between learning new tasks and consolidating the original ability. Together, these findings offer additional insights into RaDis' effectiveness and future study.

In summary, this work makes the following contributions:

- It addresses the problem of CL for instruction-tuned LLMs and discovers that LLMs can generate rationales that tie previous knowledge with new tasks without explicit prompting. Replaying these rationales effectively mitigates forgetting in a self-distillation manner.

- It proposes RaDis, a novel training approach that enhances LLMs' translation proficiency while preserving their generality by distilling self-generated rationales. Compared to the existing multi-task training approach, RaDis can inherit strong general capabilities while achieving comparable translation performance.

- Our findings provide valuable insights for developing more flexible and powerful LLMs that excel in specialized tasks without compromising their generality or safety. Additionally, the utilization of rationales to alleviate forgetting in RaDis provides a fresh angel in the field of CL.

## 2 RELATED WORKS

### 2.1 FINE-TUNING LLMS FOR MT

Previous studies have primarily fine-tuned LLMs using parallel corpora to enhance their translation proficiency. BigTrans (Yang et al., 2023) and ALMA (Xu et al., 2024a) propose to first continual pre-train on monolingual data and then progresses to fine-tuning on parallel data. Although these

methods enhance the translation proficiency of LLMs, they often compromise the models' general ability, turning them into specialized translation models. To address this issue, several studies have sought to preserve the general ability of LLMs while fine-tuning them for MT. For example, ParroT (Jiao et al., 2023), BayLing (Zhang et al., 2023) and TowerInstruct (Alves et al., 2024) incorporates general instruction-following data to maintain general capability. However, limited by the quality of the data, their performance in both translation and general abilities remains relatively low. In contrast to these efforts, our approach solely utilizes machine translation data and preserves the general ability by distillation of self-generated rationales.

## 2.2 CONTINUAL INSTRUCTION TUNING

Continual instruction tuning (CIT) seeks to mitigate CF during the instruction tuning of LLMs by employing CL approaches (Wu et al., 2024; Shi et al., 2024). Traditional CL methods are typically divided into replay-based, regularization-based, and architecture-based methods. However, in the context of LLMs, the vast parameter and task space reduces the feasibility of regularization-based and architecture-based methods (Wang et al., 2024). As a result, current research has predominantly relied on focused on replay-based techniques and their variants (Scialom et al., 2022; Yin et al., 2022; Mok et al., 2023; He et al., 2024; Wang et al., 2024) While these approaches are promising, they are subject to the reliance on access to the original training data. Consequently, they cannot be applied to mitigate the forgetting of instruction-tuned LLMs' general abilities gained from in-house training data. SDFT (Yang et al., 2024) is the first work designed for preserving the general instruction-following abilities of LLMs. It proposes to paraphrase the original train dataset with the LLM itself to bridge the distribution gap. However, the quality of the paraphrased data is limited by the capabilities of the prompt and the model itself, which may diminish the performance of the task to be learned. In contrast, RaDis argues the original data with self-generated rationales and avoids loss of performance on new tasks.

## 2.3 DISTILLING RATIONALES

Since the advent of LLMs, researchers have recognized their ability to generate rationales and have sought to distill knowledge from them. Initial studies have predominately focused on distilling the Chain of Thought (CoT) reasoning capabilities from intermediate rationales (Wang et al., 2023; Hsieh et al., 2023; Fu et al., 2023). These studies emphasize *pre-rationalization*, where the model first generates a rationale before predicting the answer based on that rationale. Here, the rationale reflects the reasoning path leading to the final answer, providing valuable insights for student models. Recent research has proposed *post-rationalization*, where the rationale is generated after the answer is predicted. In this context, the rationale serves as an explanation, supplementing the ground truth label. Wadhwa et al. (2024) demonstrate that CoT-augmented distillation is more effective when rationales are provided after labels. Additionally, Chen et al. (2024) suggests that *post-rationalization* mitigates rationale sensitivity issues and enhances focus on learning challenging samples. RationaleCL (Xiong et al., 2023) introduce rationales generated by GPT-3.5-turbo to distill relation extraction knowledge into T5 model in a continual learning setting. Unlike previous works that distill knowledge from LLMs to smaller student models, RaDis focuses on self-distillation to maintain general abilities and prevent forgetting.

## 3 METHOD

We begin by presenting a key observation: when tasked with translation requests, instruction-tuned LLMs can generate detailed **rationales** that encapsulate the internal general knowledge leveraged during translation (Section 3.1). Building on this insight, we introduce **Ra**tionale **Dis**tillation (RaDis), which leverages these self-generated rationales as replay data to help the model retain its broad general capabilities (Section 3.2). Finally, we demonstrate that the RaDis training objective can be decomposed into a conventional MT loss and a self-distillation loss on rationale tokens, which helps prevent excessive deviation of model parameters (Section 3.3).

## 3.1 OBSERVATION: SELF-GENERATED RATIONALES

Instruction-tuned LLMs exhibit a strong ability to follow instructions and engage in conversational interactions, delivering helpful responses across a wide range of tasks. Unlike conventional models that simply output the answer, instruction-tuned LLMs are known to be able to generate rationales (Wei et al., 2022). As illustrated in Figure 2, when presented with a translation request, instruction-tuned LLMs not only generate the translation but also provide an accompanying rationale. For 139 out of 200 randomly sampled translation instructions, LLaMA-2-7B-Chat provided a translation along with a rationale. In line with findings in CoT-augmented distillation (Wadhwa et al., 2024; Xu et al., 2024b), we found these rationales encompass a wealth of diverse information leveraged during translation, including word or phrase translations, sentence structure

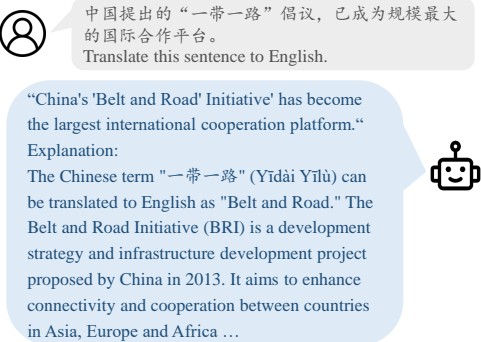

Figure 2: An example of LLM's response to translation instruction. In this case, the LLM provides a rationale with additional factual information about the term 'Belt and Road' after the translation result.

analysis, factual information about the sentence, explanations of the overall sentence meaning, and other details.

### 3.2 RaDis: distilling rationales to alleviate forgetting

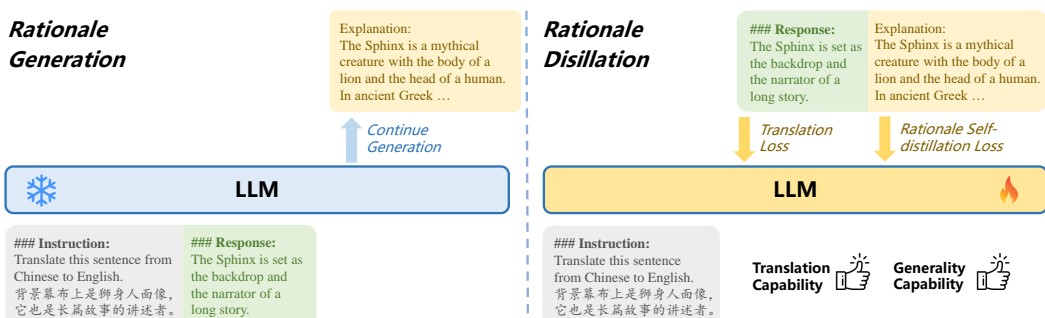

Figure 3: Overview of the RaDis approach. **Rationale Generating (Left)**: Given a translation instruction-response pair as an input, the LLM extends the response by generating a rationale. **Fine-tuning with Rationale Distillation (Right)**: RaDis utilizes this self-generated rationale to enrich the original response and fine-tunes the LLM with the enriched data. The CLM loss computed on the rationale serves as a self-distillation regularization term, preventing excessive parameter divergence.

The forgetting issue can be attributed to an unsuitable training approach. In conventional fine-tuning, the supervision signal comes from the reference sentence solely, which contains knowledge specific to the translation task. Thus, the model parameter is biased to a translation-specific distribution. Previous studies have sought to address this issue by replay-based CL methods. However, due to the absence of original training data and the poor quality of open-sourced instruction-following data, their effectiveness is limited. To this end, we propose RaDis.

The core idea of RaDis is similar to *pseudo-replay*. In traditional CL, pseudo-replay methods employ an additional data generator to synthesize replay data (Shi et al., 2024). However, the superior generative abilities of LLMs now allow us to *leverage the model itself to synthesize this replay data*. As depicted in Figure 3, RaDis starts from an instruction-tuned LLM as the backbone. It utilizes a prompt template $\mathcal{I}$ to format the translation sentence pair $(x, y)$ and sends them into the backbone LLM parameterized as $\theta$. As shown in Section 3.1, an instruction-tuned model has the inherent ability to continue generating a rationale using the translation instruction-response pair as the prefix.

$$\mathbf{r} \sim P(\mathbf{y}, \mathbf{x}, \mathcal{I}; \theta) \qquad (1)$$

These rationales encapsulate the internal general knowledge leveraged during translation, building a reasoning framework that ties previous knowledge with new tasks. Specifically, the self-generated rationale $\mathbf{r}$ is concatenated with the translation sentence $\mathbf{y}$, creating an enriched response $\hat{\mathbf{y}} = \text{CONCAT}(\mathbf{y}, \mathbf{r})$. The enriched instruction-response pair is subsequently used to train the backbone LLM using a standard causal language model (CLM) loss, defined as:

$$\mathcal{L}(\mathbf{x}, \hat{\mathbf{y}}; \theta) = -\log P(\hat{\mathbf{y}}|\mathbf{x}, \mathcal{I}; \theta) \tag{2}$$

The enriched response now incorporates both the task-specific knowledge for translation and the diverse, original knowledge embedded within the self-generated rationale. As a result, fine-tuning the model with it can learn the translation task and consolidate the original general ability simultaneously.

### 3.3 WHY RaDis WORKS: A KNOWLEDGE DISTILLATION PERSPECTIVE

In previous sections, we discovered that self-generated rationales are effective substitutes for replay data. However, since they are neither traditional replay data nor pseudo-replay data, a natural question arises: why do they work? Here, we demonstrate that RaDis can be understood as a form of knowledge distillation, a technique proven to mitigate forgetting. To explain this, we paraphrase Equation 2 as:

$$\begin{aligned}
\mathcal{L}(\mathbf{x}, \hat{\mathbf{y}}; \theta) &= -\log P(\hat{\mathbf{y}}|\mathbf{x}, \mathcal{I}; \theta) \\
&= -\sum_{t=1}^{T+R} \log P(\hat{\mathbf{y}}_t|\hat{\mathbf{y}}_{<t}, \mathbf{x}, \mathcal{I}; \theta)
\end{aligned} \tag{3}$$

where $R$ is the length of the rationale $\mathbf{r}$. The properties of the CLM loss allow us to split and reassemble the loss across each token. By separating the loss of the reference translation from the self-generated rationale, we obtain:

$$\begin{aligned}
\mathcal{L}(\mathbf{x}, \hat{\mathbf{y}}; \theta) &= -\sum_{t=1}^{T+R} \log P(\hat{\mathbf{y}}_t|\hat{\mathbf{y}}_{<t}, \mathbf{x}, \mathcal{I}; \theta) \\
&= -\sum_{t=1}^{T} \log P(\mathbf{y}_t|\mathbf{y}_{<t}, \mathbf{x}, \mathcal{I}; \theta) - \sum_{t=T+1}^{T+R} \log P(\mathbf{r}_t|\mathbf{r}_{<t}, \mathbf{y}, \mathbf{x}, \mathcal{I}; \theta) \\
&= -\log P(\mathbf{y}|\mathbf{x}, \mathcal{I}; \theta) - \log P(\mathbf{r}|\mathbf{y}, \mathbf{x}, \mathcal{I}; \theta)
\end{aligned} \tag{4}$$

Here, the first term is a traditional MT loss, which trains the model to acquire new translation knowledge. The second term minimizes the negative log-likelihood of the self-generated rationale $\mathbf{r}$ given the conventional translation instruction-response pair. It can be interpreted as a sequence-level self-distillation loss on the rationale tokens, which serves as a regularizer to mitigate forgetting by preventing excessive deviation of model parameters.

## 4 EXPERIMENTS

### 4.1 DATASETS

The datasets and benchmarks used for fine-tuning and evaluation are listed below:

**Translation.** For parallel training data, we adopt the human written data collected by ALMA (Xu et al., 2024a), as it has been proven effective in enhancing the translation proficiency of LLMs. This data comprises human written test datasets from WMT'17 to WMT'20, plus the development and test sets from Flores-200 (Goyal et al., 2022). It covers 4 English-centric language pairs, considering both from English and to English directions: Czech (`cs`), Chinese (`zh`), German (`de`), and Russian (`ru`). The WMT'22 test dataset for the same 8 translation directions is used for testing. Translation performance is evaluated using the COMET metric (`Unbabel/wmt22-comet-da`) due to its better alignment with human evaluations (Rei et al., 2022).

**Conversation and Instruction Following.** MT-Bench (Zheng et al., 2023) and AlpacaEval (Dubois et al., 2024) are employed to evaluate the conversation and instruction-following abilities of the models. MT-Bench consists of a set of challenging multi-turn questions across various categories, including math, coding, role-play, and writing. GPT-4 is utilized as the judge to assess the quality of the models' responses, scoring them on a scale of 1 to 10, as outlined by Zheng et al. (2023). The AlpacaEval and AlpacaEval 2.0 leaderboard evaluates the models on 805 prompts from the AlpacaEval dataset and calculates the win rate against text-davinci-003 and GPT-4-1106. For this evaluation, we use the `weighted_alpaca_eval_gpt4_turbo` annotator as the judge.

**Safety.** Safety is evaluated using harmful behavior datasets consisting of unsafe prompts. Following WalledEval (Gupta et al., 2024), we feed 520 unsafe prompts from AdvBench (Zou et al., 2023) into the LLMs and utilize `LLaMA-3-Guard-8B` (Dubey et al., 2024) to assess whether the responses are harmful. We report the safe rate, defined as the percentage of safe responses across all prompts.

**Reasoning.** The reasoning ability is evaluated using GSM8K (Cobbe et al., 2021), which comprises 8.8k high-quality arithmetic word problems designed at the grade school level, to assess the arithmetic reasoning abilities of LLMs. The evaluations are conducted using `lm-evaluation-harness` (Gao et al., 2024) and the exact match scores are reported.

## 4.2 BASELINES

Our method is compared against two baseline categories. The first category includes representative continual instruction tuning (CIT) methods, that are compatible with our setting. In the second category, we consider prior LLM-based MT models, which focus on enhancing the translation proficiency of LLMs. It's worth noting that the comparison with prior LLM-based MT models is not entirely fair due to discrepancies in training data and model architectures.

**Continual Instruction Tuning (CIT) Baselines.** The following continual instruction tuning approaches are introduced as baseline methods: **(1) Vanilla Fine-tuning**, where the backbone LLMs are directly fine-tuned using translation data; **(2) Seq-KD**, which employs sequence-level knowledge distillation along with fine-tuning to alleviate forgetting; **(3) SDFT** (Yang et al., 2024), which leverages the backbone LLM to paraphrase the original training data and fine-tunes the model using the synthesized data; **(4) Multi-task**, which employs open-sourced instruction following dataset and fine-tunes the LLM with both translation and instruction following data.

**Prior LLM-based MT models.** This category contains several notable works in the field of LLM-based machine translation. **(1) ParroT** (Jiao et al., 2023), which fine-tune LLaMA-1 with a hybrid of translation and instruction-following data. **(2) BigTrans** (Yang et al., 2023) enhances LLaMA-1 by equipping it with multilingual translation capabilities across more than 100 languages. **(3) BayLing** (Zhang et al., 2023) fine-tunes LLaMA-1 using automatically generated interactive translation instructions. **(4) ALMA** (Xu et al., 2024a) first fine-tunes LLaMA-2 on monolingual data and subsequently uses high-quality parallel data for instruction tuning. **(5) TowerInstruct (Alves et al., 2024)** continued pre-trains LLaMA-2 on a multilingual mixture of monolingual and parallel data, and fine-tuned with translation and instruction following data.

Please refer to Appendix A for further details of the baselines.

## 4.3 TRAINING DETAILS

In our experiments, we employ LLaMA-2-7B-Chat (Touvron et al., 2023) and Mistral-7B-Instruct-v0.2 (Jiang et al., 2023) as the backbone LLMs. Given the constraints of our computational resources, the Low-Rank Adaptation (LoRA) technique (Hu et al., 2022) is utilized in most of our experiments. Specifically, a LoRA adapter with a rank of 16 is integrated into all the linear layers of the LLMs and exclusively trains the adapter. The LLMs are fine-tuned for three epochs on the translation dataset, with a learning rate of $1 \times 10^{-4}$ and a cosine annealing schedule. The batch is set to 128 for stable training. Our implementation is based on `LLaMA-Factory` (Zheng et al., 2024). After the fine-tuning phase, the LoRA module is merged into the backbone LLM for testing. For further details, please refer to Appendix B.

## 4.4 RESULTS

Table 1: The overall translation performance (COMET score) in EN→X. The delta performance compared to the backbone LLM is shown.

| Models | Czech | German | Russian | Chinese | Avg. |
|---|---|---|---|---|---|
| | *Backbone LLM: LLaMA-2-7B-Chat* | | | | |
| **Backbone LLM** | 70.14 | 75.10 | 75.76 | 72.57 | 73.39 |
| **w/ Vanilla-FT** | 81.80 ↑ 11.66 | 82.81 ↑ 7.71 | 84.67 ↑ 8.91 | 81.96 ↑ 9.39 | 82.81 ↑ 9.42 |
| **w/ Multi-task** | 81.67 ↑ 11.53 | 82.58 ↑ 7.48 | 84.24 ↑ 8.48 | 81.86 ↑ 9.29 | 82.59 ↑ 9.20 |
| **w/ Seq-KD** | 70.17 ↑ 0.03 | 74.40 ↓ 0.7 | 75.62 ↓ 0.14 | 72.93 ↑ 0.36 | 73.28 ↓ 0.11 |
| **w/ SDFT** | 68.59 ↓ 1.55 | 75.21 ↑ 0.11 | 79.67 ↑ 3.91 | 78.45 ↑ 5.88 | 75.48 ↑ 2.09 |
| **w/ RaDis (Ours)** | 81.77 ↑ 11.63 | 82.39 ↑ 7.29 | 84.31 ↑ 8.55 | 81.98 ↑ 9.41 | 82.61 ↑ 9.22 |
| | *Backbone LLM: Mistral-7B-Instruct-v0.2* | | | | |
| **Backbone LLM** | 67.39 | 67.87 | 64.56 | 71.32 | 67.79 |
| **w/ Vanilla-FT** | 84.33 ↑ 16.94 | 83.04 ↑ 15.17 | 86.23 ↑ 21.67 | 83.63 ↑ 12.31 | 84.31 ↑ 16.52 |
| **w/ Multi-task** | 84.79 ↑ 17.40 | 82.64 ↑ 14.77 | 86.47 ↑ 21.91 | 83.87 ↑ 12.55 | 84.44 ↑ 16.56 |
| **w/ Seq-KD** | 74.30 ↑ 6.91 | 73.69 ↑ 5.82 | 73.10 ↑ 8.54 | 78.28 ↑ 6.96 | 74.84 ↑ 7.06 |
| **w/ SDFT** | 51.90 ↓ 15.49 | 53.32 ↓ 14.55 | 47.07 ↓ 17.49 | 56.38 ↓ 14.94 | 52.17 ↓ 15.62 |
| **w/ RaDis (Ours)** | 84.32 ↑ 16.93 | 82.95 ↑ 15.08 | 86.55 ↑ 21.99 | 83.75 ↑ 12.43 | 84.39 ↑ 16.61 |
| | *Prior LLM-based MT Models* | | | | |
| **ParroT** | - | 81.20 | - | 79.30 | - |
| **BigTrans** | 80.65 | 78.81 | 78.21 | 81.31 | 79.75 |
| **BayLing-7B** | 76.85 | 82.18 | 74.72 | 84.43 | 79.55 |
| **ALMA-7B** | 89.05 | 85.45 | 87.05 | 84.87 | 86.61 |

Table 2: The overall translation performance (COMET score) in X→EN. The delta performance compared to the backbone LLM is shown.

| Models | Czech | German | Russian | Chinese | Avg. |
|---|---|---|---|---|---|
| | *Backbone LLM: LLaMA-2-7B-chat* | | | | |
| **Backbone LLM** | 79.53 | 81.20 | 80.36 | 74.95 | 79.01 |
| **w/ Vanilla-FT** | 82.74 ↑ 3.21 | 83.31 ↑ 2.11 | 82.70 ↑ 2.34 | 78.08 ↑ 3.13 | 81.71 ↑ 2.7 |
| **w/ Multi-task** | 82.71 ↑ 3.18 | 83.37 ↑ 2.17 | 82.73 ↑ 2.37 | 78.20 ↑ 3.25 | 81.75 ↑ 2.74 |
| **w/ Seq-KD** | 78.50 ↓ 1.03 | 80.61 ↓ 0.59 | 79.88 ↓ 0.48 | 74.48 ↓ 0.47 | 78.37 ↓ 0.64 |
| **w/ SDFT** | 81.93 ↑ 2.4 | 82.60 ↑ 1.4 | 81.87 ↑ 1.51 | 76.77 ↑ 1.82 | 80.79 ↑ 1.78 |
| **w/ RaDis (Ours)** | 81.75 ↑ 2.22 | 83.07 ↑ 1.87 | 82.22 ↑ 1.86 | 77.83 ↑ 2.88 | 81.22 ↑ 2.21 |
| | *Backbone LLM: Mistral-7B-Instruct-v0.2* | | | | |
| **Backbone LLM** | 81.88 | 81.73 | 81.97 | 77.76 | 80.84 |
| **w/ Vanilla-FT** | 83.51 ↑ 1.63 | 83.35 ↑ 1.62 | 83.23 ↑ 1.26 | 79.21 ↑ 1.45 | 82.33 ↑ 1.49 |
| **w/ Multi-task** | 82.84 ↑ 0.96 | 83.04 ↑ 1.31 | 83.15 ↑ 1.18 | 79.31 ↑ 1.55 | 82.09 ↑ 1.25 |
| **w/ Seq-KD** | 81.66 ↓ 0.22 | 81.98 ↑ 0.25 | 82.50 ↑ 0.53 | 77.49 ↓ 0.27 | 80.91 ↑ 0.07 |
| **w/ SDFT** | 80.38 ↓ 1.5 | 79.72 ↓ 2.01 | 80.21 ↓ 1.76 | 77.39 ↓ 0.37 | 79.43 ↓ 1.41 |
| **w/ RaDis (Ours)** | 82.41 ↑ 0.53 | 82.86 ↑ 1.13 | 83.32 ↑ 1.35 | 79.17 ↑ 1.41 | 81.94 ↑ 1.1 |
| | *Prior LLM-based MT Models* | | | | |
| **ParroT** | - | 82.40 | - | 75.20 | - |
| **BigTrans** | 81.19 | 80.68 | 77.80 | 74.26 | 78.48 |
| **BayLing-7B** | 82.03 | 83.19 | 82.48 | 77.48 | 81.30 |
| **ALMA-7B** | 85.93 | 83.95 | 84.84 | 79.78 | 83.63 |

The translation performance in EN→X and X→EN are shown in Table 1 and Table 2, respectively. The performance on general ability, including instruction following, safety, and reasoning, is shown in Table 3.

**Fine-tuning is a double-edged sword.** In the EN→X direction, Vanilla-FT significantly enhances translation performance compared to zero-shot results, achieving an average COMET score improvement of **+16.52**. In the X→EN direction, the performance improvement is relatively small (**+1.49** COMET). This is mainly because the backbone LLMs already have a strong ability to translate to English. However, this improvement in translation proficiency comes at the cost of a substantial decline in general capabilities, as reflected by the sharp performance drop in instruction-following, safety, and reasoning benchmarks.

**RaDis balances translation proficiency and general abilities.** Multi-task achieves performance comparable to Vanilla-FT. However, its performance on general tasks declines significantly, despite the inclusion of additional instruction-following data. This is because the external instruction data

Table 3: The performance on instruction following, safety, and reasoning benchmarks. **RP** represents relative performance compared to the backbone LLM and is only calculated for CIT methods. The delta performance compared to vanilla fine-tuning (Vanilla-FT) is shown. The safety rate for **BigTrans** and **ALMA** is omitted, as these translation-specific models can not generate reasonable responses.

| Models | Conversation and Instruction Following | | | Safety | Reasoning | RP[%] |
|---|---|---|---|---|---|---|
| | MT-bench | AlpacaEval | AlpacaEval 2.0 | AdvBench | GSM8K | |
| *Backbone LLM: LLaMA-2-7B-chat* | | | | | | |
| Backbone LLM | 6.51 | 71.40 | 9.66 | 100.00 | 21.83 | - |
| w/ Vanilla-FT | 1.5 | 2.18 | 0.71 | 37.88 | 4.32 | 18.22 |
| w/ Multi-task | 5.64 ↑ 4.14 | 44.55 ↑ 42.37 | 3.98 ↑ 3.27 | 98.65 ↑ 60.77 | 11.98 ↑ 7.66 | 68.75 ↑ 50.53 |
| w/ Seq-KD | 6.59 ↑ 5.09 | 67.48 ↑ 65.30 | 8.33 ↑ 7.62 | 100.00 ↑ 62.12 | 19.48 ↑ 15.16 | 94.24 ↑ 76.02 |
| w/ SDFT | 5.66 ↑ 4.16 | 67.55 ↑ 65.37 | 7.09 ↑ 6.38 | 98.08 ↑ 60.20 | 20.02 ↑ 15.70 | 88.95 ↑ 70.72 |
| w/ RaDis | 6.56 ↑ 5.06 | 67.94 ↑ 65.76 | 7.47 ↑ 6.76 | 100.00 ↑ 62.12 | 19.48 ↑ 15.16 | 92.50 ↑ 74.27 |
| *Backbone LLM: Mistral-7B-Instruct-v0.2* | | | | | | |
| Backbone LLM | 7.67 | 84.91 | 15.09 | 68.46 | 41.62 | - |
| w/ Vanilla-FT | 1.94 | 6.07 | 1.02 | 4.23 | 0.23 | 9.19 |
| w/ Multi-task | 6.87 ↑ 4.93 | 49.46 ↑ 43.39 | 5.45 ↑ 4.43 | 63.85 ↑ 59.62 | 22.97 ↑ 22.74 | 66.48 ↑ 57.29 |
| w/ Seq-KD | 6.99 ↑ 5.05 | 82.06 ↑ 75.99 | 12.7 ↑ 11.68 | 60.58 ↑ 56.35 | 41.77 ↑ 41.54 | 92.16 ↑ 82.97 |
| w/ SDFT | 7.00 ↑ 5.06 | 78.32 ↑ 72.25 | 10.02 ↑ 9.00 | 48.27 ↑ 44.04 | 41.09 ↑ 40.86 | 83.83 ↑ 74.64 |
| w/ RaDis | 7.57 ↑ 5.63 | 80.34 ↑ 74.27 | 11.05 ↑ 10.03 | 62.12 ↑ 57.89 | 41.70 ↑ 41.47 | 91.49 ↑ 82.31 |
| *Prior LLM-based MT models* | | | | | | |
| ParroT | 4.58 | 28.06 | 2.82 | 26.35 | 4.09 | - |
| BigTrans | 1.73 | 0.42 | 0.22 | - | 0 | - |
| BayLing-7B | 4.51 | 51.29 | 4.43 | 84.62 | 5.38 | - |
| ALMA-7B | 2.80 | 1.08 | 0.17 | - | 0 | - |

is of low quality and out-of-distribution relative to the backbone LLM. As a result, fine-tuning these data does not alleviate the issue of catastrophic forgetting. The translation results of SDFT fall below the zero-shot performance of backbone LLM when using Mistral-7B-Instruct-v0.2 as the backbone LLM. This under-performance stems from the fact that the prompt used for rewriting data is tailored for LLaMA-2 models, which does not generalize well to Mistral models. Additionally, SDFT shows weaker performance in EN→X translations compared to X→EN, indicating that the limited ability of the backbone LLM to translate into other languages diminishes the quality of the distilled dataset. Seq-KD preserves up to 94.24% of the overall general capabilities but brings almost no improvement in translation performance. In contrast, RaDis strikes a better balance between translation proficiency and general ability. It achieves a COMET score comparable to Vanilla-FT (81.58 vs. 82.02, 83.17 vs. 83.56) while preserving up to 92.50% of the general capabilities.

**Compared with prior LLM-based MT models.** RaDis helps backbone models perform comparably with SOTA LLM-based MT models in translation performance. Our best model (Mistral-7B-Instruct-v0.2 w/ RaDis) surpasses BigTrans, BayLing, and ParroT by a considerable margin in average COMET scores, particularly in the EN→X direction. Although ALMA achieves a higher COMET score than our best model, it benefits from continual pre-training on massive monolingual data, which is not implemented in our approach. Regarding general ability, models equipped with RaDis significantly outperform all prior studies. Translate-specific models, such as BigTrans and ALMA that are only fine-tuned with translation data, lack general ability. While ParroT and BayLing utilize Alpaca data in training, their general ability is limited by their backbone models and the quality of Alpaca data. In contrast, RaDis preserves the strong general ability of the backbone LLMs, achieving the highest performance. As TowerInstruct has been fine-tuned on the WMT'22 test dataset, we evaluate translation performance on the WMT'23 test dataset following their setting. The detailed comparison can be found in Appendix C.

## 5 ANALYSIS

### 5.1 DO RATIONALES CONTAIN GENERAL KNOWLEDGE?

To investigate the content of self-generated rationales, we randomly sampled 25 instances from each translation direction, forming a set of 200 for analysis. The content of the rationales included diverse information, as expected. As shown in Table 4, the information can be broadly categorized into eight

Table 4: Rationale categories and their main contents. Note that the percentages do not sum to 100%, as each rationale may include multiple categories of information.

| Information Category | Contents | Occurrence [%] |
|---|---|---|
| Word/phrase translation | Translations of individual words and phrases in the sentence. | 29.5 |
| Alternative translation | Multiple translation options provided for selection. | 19.5 |
| Helpful&Safety | Guidance on helpful and safe practices when translating sensitive sentences. | 17.5 |
| Semantic explanation | Clarification of the sentence's meaning. | 15.5 |
| Back-translation | Back-translations of non-English results to verify their accuracy. | 12.5 |
| Factual supplement | Additional factual information regarding entities or events in the sentence. | 9.0 |
| Word/phrase explanation | Explanations for special words, idioms, or expressions. | 7.5 |
| Grammar | Information on grammar, such as part-of-speech or sentence structure. | 5.5 |

types, ranging from word alignments to factual knowledge. These rationales act as a "semantic scaffold", linking old knowledge to new tasks and building an underlying reasoning framework that ties everything together, which improves knowledge retention. For examples of rationales, please refer to Appendix E.

## 5.2 WHICH IS THE KEY: RATIONALE QUALITY OR SELF-DISTILLATION PROPERTY?

Table 5: The result of ablation study. The names in brackets are the models used to generate rationales. The best results in different RaDis variants are highlighted in bold.

| Models | Machine Translation | | Conversation and Instruction Following | | | Safety | Reasoning |
|---|---|---|---|---|---|---|---|
| | X→EN | EN→X | MT-bench | AlpacaEval | AlpacaEval 2.0 | AdvBench | GSM8K |
| **Mistral-7B-Instruct-v0.2** | 80.84 | 67.79 | 7.67 | 84.91 | 15.09 | 68.46 | 41.62 |
| **w/ Vanilla-FT** | 82.33 | 84.31 | 1.94 | 6.07 | 1.02 | 4.23 | 0.23 |
| **w/ RaDis (Self-generated)** | 81.94 | 84.39 | **7.57** | **80.34** | **11.05** | 62.12 | **41.70** |
| **w/ RaDis (LLaMA-2-Chat-7B)** | 82.33 | 84.51 | 6.89 | 63.85 | 6.28 | **98.46** | 35.03 |
| **w/ RaDis (LLaMA-3-70B-Instruct)** | **82.84** | **84.71** | 7.00 | 77.41 | 10.27 | 58.27 | 39.42 |

The effectiveness of RaDis can be explained in two ways: rationale quality (in terms of the knowledge they contain) and the self-distillation property. We conducted the following ablation experiment to analyze the impact of these two factors. Specifically, the self-generated rationales in RaDis were replaced by rationales generated by different models, namely LLaMA-2-7B-Chat and LLaMA-3-70B-Instruct (Dubey et al., 2024). Due to the difference in parameter size and fine-tuning data, these models can provide rationales with varying levels of quality, but all lack the self-distillation property. Therefore, it is possible to separate the rationale quality and the self-distillation property to analyze their contributions.

As shown in Table 5, RaDis consistently mitigates the forgetting of general capabilities, regardless of the type of rationales used. When comparing rationales generated by LLaMA-2-7B-Chat and LLaMA-3-70B-Instruct, the latter demonstrates superior performance in both MT and general tasks, except for the safety task. These results suggest that models can learn more translation knowledge from higher-quality rationales, which leads to better performance. However, self-generated rationales demonstrate the strongest ability to retain general capabilities, even outperforming those generated by LLaMA-3-70B-Instruct, highlighting the importance of the self-distillation property.

## 5.3 RADIS AVOIDS THE CONFLICT BETWEEN LEARNING AND MITIGATING CF

As demonstrated in Section 3.3, our proposed RaDis can be viewed as a specialized form of sequence-level distillation, where the rationale $\mathbf{r}$ serves as the distillation target. However, while both methods excel at preserving general capabilities, RaDis notably enhances translation proficiency, whereas Seq-KD does not. We posit that the difference arises from whether the regularization term conflicts with the MT learning process. In Seq-KD (Equation 5), the MT loss $-\log P(\mathbf{y}|\mathbf{x}, \mathcal{I}; \theta)$ and the regularization term $-\log P(\mathbf{y}'|\mathbf{x}, \mathcal{I}; \theta)$ share the same input but have different outputs, which may lead to conflict in optimization. In contrast, with RaDis (Equation 4), the MT loss and the regularization term $-\log P(\mathbf{r}|\mathbf{y}, \mathbf{x}, \mathcal{I}; \theta)$ are less likely to exhibit this issue.

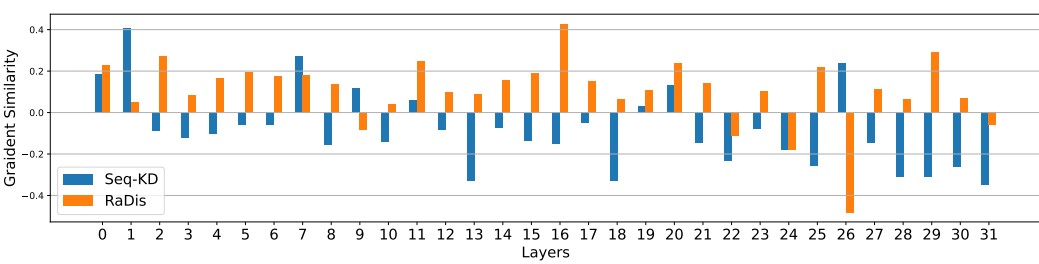

Figure 4: Overview of the gradient similarity between the regularization term and MT loss.

We analyze gradient similarity to validate our assumption. Specifically, we sample 128 examples from each translation direction to create a validation set consisting of 1024 samples. Subsequently, the gradient features for the MT loss and the regularization term for both methods are extracted, following Xia et al. (2024). Finally, the cosine similarity of the gradient features is computed. As depicted in Figure 4, in 24 out of 32 layers, the gradient of the regularization term for Seq-KD exhibits negative similarity with the MT loss, indicating a significant conflict between these objectives. In contrast, in 25 out of 32 layers, the gradient of RaDis' regularization term shows positive similarity with the MT loss, suggesting that RaDis can avoid the conflict between learning and mitigating forgetting.

## 6   CONCLUSIONS

In this paper, we conducted a systematic evaluation of prior LLM-based MT models, finding that they lack diverse general capabilities, degenerating into task-specific translation models. This degeneration is mainly caused by catastrophic forgetting while fine-tuning for translation tasks. Previous continual learning approaches can not preserve the general capabilities gained from in-house training data. To address this issue, we propose a simple yet effective strategy, RaDis. RaDis prompts LLMs to generate rationales for the reference translation and utilizes these rationales to mitigate forgetting in a self-distillation manner. Mirroring the human learning process, these rationales connect prior knowledge with new tasks, building a reasoning framework that ties internal concepts together and enhances knowledge retention. Extensive experiments show that RaDis greatly enhances the translation performance while preserving the models' general ability. These insights can help future research on building LLMs capable of excelling in specialized tasks without compromising their generality or safety and providing a fresh angle for utilizing rationales in the CL field.

## 7   LIMITATIONS AND FUTURE WORK

Our study is subject to certain limitations. Owing to constraints in computational resources, we adopt LoRA on models with 7B parameters. Further investigations involving larger models and full fine-tuning remain to be explored. Besides, as a post-training method, RaDis is limited by the language proficiency of backbone LLM. This limits its performance on low-resource language. However, we believe the rapidly evolving multilingual LLMs would narrow this gap. Furthermore, we predominately focus on fine-tuning with machine translation data, applying RaDis to other NLP tasks will further support its effectiveness (See Appendix C). This potential direction is what we intend to explore in future work.

### REPRODUCIBILITY STATEMENT

Codes and model weights will be made public after review to advocate future research. For synthesizing data, we provide several examples in Appendix E. For evaluation, we primarily use greedy decoding to ensure reproducibility, except where specific generation configurations are mandated by certain benchmark tools. Note that evaluations on instruction-following abilities (AlpacaEval and MT-Bench) rely on OpenAI's API. The randomness of API responses may have little impact on the reproducibility of these benchmarks.

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

# A BASELINE DETAILS

## A.1 CONTINUAL INSTRUCTION TUNING BASELINES

**Vanilla Fine-tuning.** This method directly fine-tunes the backbone LLMs with translation data, without incorporating any mechanism to address the forgetting issue.

**Sequence-level Knowledge Distillation (Seq-KD).** This method first sends the formatted translation instructions to the backbone LLM and generates outputs $\mathbf{y}'$. The model is trained with both golden references $\mathbf{y}$ and the self-generated outputs $\mathbf{y}'$. The overall training objective is:

$$
\begin{aligned}
\mathcal{L}_{\text{Seq-KD}} =& \mathcal{L}(\mathbf{x}, \mathbf{y}; \theta) + \mathcal{L}(\mathbf{x}, \mathbf{y}'; \theta) \\
=& -\log P(\mathbf{y}|\mathbf{x}, \mathcal{I}; \theta) - \log P(\mathbf{y}'|\mathbf{x}, \mathcal{I}; \theta)
\end{aligned}
\tag{5}
$$

**Self-distillation Fine-tuning (SDFT). (Yang et al., 2024)** This method first prompts the backbone LLM to paraphrase the original responses present in the task dataset, yielding a distilled dataset. Subsequently, the distilled dataset, which is used in subsequent fine-tuning, helps narrow the distribution gap between LLM and the original dataset. We adopt the general distillation template provided in their paper to paraphrase the dataset.

**Multi-task fine-tuning (Multi-task)** . This method employs open-sourced instruction following datasets and fine-tunes the LLM with both translation and instruction following data. Specifically, we adopt Alpaca (Taori et al., 2023) and Dolly (Conover et al., 2023) as the chosen instruction following dataset. Note that multi-task fine-tuning utilizes more data in the training process and is usually considered the upper bound of the continual learning approaches.

## A.2 PRIOR LLM-BASED MT MODELS

**ParroT** (Jiao et al., 2023) reformulates translation data into the instruction-following style and introduces a "Hint" field for incorporating extra requirements to regulate the translation process. It is also fine-tuned on the Alpaca dataset to enhance general ability.

**BigTrans** (Yang et al., 2023) continual pre-train LLaMA-1-13B with Chinese monolingual data and an extensive parallel dataset encompassing 102 natural languages. They then apply multilingual translation instructions for fine-tuning.

**BayLing** (Zhang et al., 2023) builds an interactive translation dataset and fine-tune LLaMA-1 models with both interactive translation and instruction-following datasets (Alpaca).

**ALMA** (Xu et al., 2024a) proposes a new training recipe for building LLM-based MT models, which begins with initial fine-tuning on monolingual data and then progresses to fine-tuning on a select set of high-quality parallel data.

**TowerInstruct** (Alves et al., 2024) propose a recipe for tailoring LLMs to multiple tasks present in translation workflows. They perform continued pre-training on a multilingual mixture of monolingual and parallel data, followed by fine-tuning instructions relevant to translation processes and general tasks.

In our experiments, we utilized the 7B models as baselines for ParroT, BayLing, ALMA and TowerInstruct to ensure a fair comparison of model size.

# B  TRAINING DETAILS

## B.1  PROMPT TEMPLATES

In all experiments, we use the original instruction format of the backbone LLM for both rationale generation and fine-tuning. For LLaMA To avoid the overfit on specific instructions. 5 different translation instructions are generated and randomly applied to each sample. The instructions are shown in Figure 5.

---

**Instruction 1:**
Could you please translate this sentence from {lang1} to {lang2}?
{sent1}
**Instruction 2:**
Translate the following sentence from {lang1} to {lang2}:
{sent1}
**Instruction 3:**
Translate this sentence from {lang1} to {lang2}.
{sent1}
**Instruction 4:**
Translate from {lang1} to {lang2}:
{sent1}
**Instruction 5:**
{sent1}
Translate this sentence to {lang2}.

---

Figure 5: The translation instructions.

## B.2  HYPERPARAMETER

Due to the limitation of resources, our experiments utilize the Low-Rank Adaptation (LoRA) technique (Hu et al., 2022). Specifically, we integrate a LoRA adapter with a rank of 16 into all the linear layers of the LLMs and exclusively train the adapter. The LLMs are fine-tuned over three epochs on the translation dataset, which equates to approximately 2,500 steps. We use a learning rate of $1 \times 10^{-4}$ and a batch size of 128 to ensure stable training across most experiments. An exception is Seq-KD, which requires a batch size of 256 to maintain the same number of training steps. All experiments are performed on 4 NVIDIA A100 80GB GPUs. For data synthesis, we employ vllm (Kwon et al., 2023) to facilitate fast data generation. For evaluation, we primarily use greedy decoding to ensure reproducibility, except where specific generation configurations are mandated by certain benchmark tools.

# C  COMPARISON WITH TOWERINSTRUCT

Given that TowerInstruct has been fine-tuned on the WMT'22 test set, we shifted the translation test set to the WMT'23 test set. We report the performance of the best model in our paper (Mistral+RaDis) alongside ALMA and TowerInstruct. To further demonstrate the potential of our approach, we also conducted new experiments with Qwen2.5-Instruct as the backbone (Qwen2.5+RaDis).

As shown in Table 6, RaDis consistently outperforms TowerInstruct-v0.2 in terms of preserving general abilities. This is primarily due to the fact that TowerInstruct-v0.2 is fine-tuned using UltraChat, which, like other open-sourced instruction datasets, suffers from lower quality.

In terms of translation, TowerInstruct-v0.2 achieves higher performance, largely due to the benefits of multilingual pre-training and extensive parallel fine-tuning. However, we would like to emphasize the strong potential of our approach from two key perspectives:

- **RaDis is more efficient**: The training times for TowerBase 7B and 13B were 80 and 160 GPU days, respectively, using A100-80GB GPUs. Fine-tuning TowerInstruct adds an ad-

Table 6: Comparison to TowerInstruct-v0.2. The best result in each column is marked in **bold**. The second best is *italicized*.

| Models | Machine Translation | | Conversation and Instruction Following | | | Safety | Reasoning |
|---|---|---|---|---|---|---|---|
| | X→EN | EN→X | MT-bench | AlpacaEval | AlpacaEval 2.0 | AdvBench | GSM8K |
| **Mistral-7B-Instruct-v0.2** | 80.84 | 67.79 | 7.67 | 84.91 | 15.09 | 68.46 | 41.62 |
| **w/ RaDis** | 80.64 | 80.58 | 7.57 | 80.34 | 11.05 | 62.12 | 41.70 |
| **Qwen2.5-7B-Instruct** | 80.90 | 80.50 | **8.58** | **88.46** | **31.55** | **99.81** | *87.72* |
| **w/ RaDis** | *82.13* | *82.81* | *8.44* | *85.62* | *27.91* | *99.04* | **88.78** |
| ALMA-7B | 81.65 | 81.91 | 2.80 | 1.08 | 0.17 | - | 0.00 |
| TowerInstruct-7B-v0.2 | **82.77** | **84.28** | 5.71 | 51.59 | 4.02 | 30.19 | 7.35 |

> ditional 200 GPU hours. In contrast, RaDis requires only 20 GPU hours (4 hours for generating rationales and 16 hours for training), which is less than 1% of the training cost for TowerInstruct-7B, while still achieving strong performance.

> • **RaDis can benefit from stronger backbone LLM**: While TowerInstruct achieves better translation performance, RaDis can effectively bridge this gap by leveraging a stronger backbone LLM. As shown in 'Table 2', switching the backbone from Mistral to Qwen2.5 leads to substantial improvements across all tasks and outperforms ALMA. We believe that as open-source multilingual LLMs continue to improve, the performance gap in translation will gradually narrow.

Together, these results underscore the advantages of our approach and demonstrate that RaDis offers a novel and competitive paradigm for building LLMs that excel in both translation proficiency and general ability.

## D  GENERALIZING RADIS TO OTHER TASKS

In this paper, we predominately grounded RaDis to the MT task. However, RaDis can serve as a universal CIT method for broader tasks. In this section, we demonstrate this potential with the code generation task. Specifically, we fine-tuned Mistral-v0.2 on Python code data from the Magicoder dataset (Wei et al., 2024) and evaluated its performance using HumanEval (Chen et al., 2021) and general ability benchmarks.

Table 7: Experiments on code generation. The best result in each column is marked in **bold**.

| Models | Code Generation | Conversation and Instruction Following | | Safety | Reasoning |
|---|---|---|---|---|---|
| | HumanEval | AlpacaEval | AlpacaEval 2.0 | AdvBench | GSM8K |
| **Mistral-7B-Instruct-v0.2** | 36.59 | 84.91 | 15.09 | 68.46 | 41.62 |
| **w/ Vanilla-FT** | 42.07 | 73.89 | 8.75 | 40.00 | **43.97** |
| **w/ SDFT** | 40.24 | 78.58 | 10.46 | 48.08 | 40.86 |
| **w/ RaDis** | **43.90** | **80.25** | **11.4** | **51.92** | 42.91 |

As shown in Table 7, RaDis outperforms Vanilla-FT and SDFT in code generation tasks, achieving higher Pass@1 on HumanEval and excelling in other benchmarks for general abilities.

A key reason for this is that RaDis directly preserves the original references in the dataset, whereas SDFT paraphrases them. Intuitively, while paraphrasing helps bridge the distribution gap, it also reduces the amount of learnable knowledge. As a result, SDFT may struggle to outperform Vanilla-FT on certain tasks. In contrast, RaDis directly utilizes the original references, preserving all the knowledge embedded in the data.

Regarding performance on general tasks, RaDis still outperforms SDFT. We believe this can be attributed to the distribution gap. While SDFT claims to distill the dataset, it actually paraphrases the data. As a result, the model's responses are sampled from the paraphrased instruction's output distribution, which tends to be out-of-distribution relative to the original task instruction. In contrast, RaDis performs self-distillation using rationales, which are fully in-distribution. This enables RaDis to more effectively alleviate forgetting and better preserve general abilities.

These results suggest that RaDis generalizes well to a broader range of tasks, highlighting its potential as a robust, general-purpose continual instruction tuning method. We plan to investigate this potential in future works.

## E  RATIONALE EXAMPLES

Several examples of rationales generated by Mistral-7B-Instruct-v0.2 are provided in Figure 7,8,9,10,11,12,13.

---

**Instruction:**
Translate this sentence from Czech to English.
Policie celý případ řádně prošetřila a útočníka posléze vypátrala.

**Response:**
The police investigated the case thoroughly and finally tracked down the assailant.

**Rationale:**
Explanation:
In this sentence, "policie" means "police," "celý případ" means "the entire case," "řádně prošetřila" means "thoroughly investigated," and "vypátrala" means "tracked down" or "found." The past tense verbs "prošetřila" and "vypátrala" both end in "-la," which is the feminine singular past tense ending in Czech. Since there is no definite article before "policie," it is assumed to be in the neutral form, which is why the verb forms are in the feminine singular. In English, we keep the same meaning but change the word order to "the police investigated the case thoroughly and finally found the assailant."

---

Figure 6: Example of rationale containing word/phrase translation.

---

**Instruction:**
Could you please translate this sentence from Czech to English?
Tak to vypadá, když se lidé za odpočinkem a zábavou přesunou někam, kde je to velmi odlišné od jejich každodenního života.

**Response:**
This is when people go to a place that is very different from their regular day-to-day life to relax and have fun.

**Rationale:**
or
It looks like this when people move to a place that is quite different from their everyday life for relaxation and enjoyment.

---

Figure 7: Example of rationale containing alternative translation.

**Instruction:**
Translate the following sentence from English to Chinese:
But denuclearization negotiations have stalled.

**Response:**
但无核化谈判陷入了僵局。

**Rationale:**
This translation is provided for information purposes only and may not be 100% accurate. It is always recommended to consult a professional translator for important documents or conversations.

Figure 8: Example of rationale containing helpful&safety information.

**Instruction:**
Translate from English to Czech:
It will support their training.

**Response:**
Podpoří jejich výcvik.

**Rationale:**
This sentence means that the action or thing being referred to will help or contribute to the training of the people mentioned.

Figure 9: Example of rationale containing semantic explanation.

**Instruction:**
Could you please translate this sentence from English to Chinese?
According to their research, a single tea bag can release more than 11.6 billion plastic particles, "which is higher than the plastic load of other earlier foods by several magnitudes".

**Response:**
根据这项研究显示，他们发现，单一茶袋释出超过116亿颗塑料微粒，"较先前其他食物的塑料负荷量高出好几个级数"。

**Rationale:**
This sentence translates to:
According to their research, a single tea bag releases over 11.6 billion plastic particles, which is several magnitudes higher than the plastic load of other earlier foods.

Figure 10: Example of rationale containing back-translation.

**Instruction:**
Translate from English to Czech:
Its second claw was larger, giving rise to the name Hesperonychus which means "western claw."

**Response:**
Jeho druhý dráp byl větší, a proto byl pojmenován Hesperonychus, což znamená „západní dráp".

**Rationale:**
This dinosaur was a theropod, a group of dinosaurs that includes meat-eating species. Hesperonychus lived during the Late Cretaceous period, around 75 to 73 million years ago, in what is now the southwestern United States. Its fossils have been found in the Morrison Formation in Colorado and Utah.
Hesperonychus was a relatively small theropod, with an estimated length of about 3 meters (10 feet) and weight of around 30 kilograms (66 pounds). It had a long, low snout and small, sharp teeth, suggesting that it may have been an agile predator that hunted small prey.
The most distinctive feature of Hesperonychus was its second toe, which was much larger than the others. This feature gave rise to its name, as "hespero" means "western" and "nychus" means "claw." The large claw was likely used for gripping and holding onto branches or prey, making Hesperonychus a unique and intriguing dinosaur.
Despite its small size, Hesperonychus was an important discovery in the study of dinosaur evolution, as it provided evidence of the diversity of theropod foot structures during the Late Cretaceous period.

Figure 11: Example of rationale containing factual supplement.

**Instruction:**
Could you please translate this sentence from Czech to English?
„I kdyby všechno ostatní na světě bylo špatně, on byl nebroušený diamant, který ozářil každý den", řekl."

**Response:**
"With everything else that's going wrong with the world, he was that diamond in the rough that was shining bright every day," he said.

**Rationale:**
This idiomatic expression "diamond in the rough" refers to someone who has great potential or value, despite not being polished or perfect yet.

Figure 12: Example of rationale containing word/phrase explanation.

**Instruction:**
Wir können froh sein, dass wir diese Zeit längst überwunden haben.
Translate this sentence to English.

**Response:**
We can be glad that we have long overcome this time.

**Rationale:**
Explanation: The sentence in German means "We can be glad that we have got over this time a long time ago." The past perfect tense "haben überwunden" (have overcome) is used to express that an action was completed in the past. The subjunctive mood "können sein" (can be) is used to express a state of being or a condition. The past participle "überwunden" agrees with the subject "wir" (we) in gender and number. The article "diese" refers to the time mentioned earlier in the sentence. The word "längst" (long) is an adverb that modifies the verb "überwunden" and indicates that the action was completed a long time ago. The sentence in English conveys the same meaning, but uses the past perfect continuous tense "have been glad" to express the same idea of a past state that continues up to the present.

Figure 13: Example of rationale containing grammar information.

