# OpenReview forum: "Boosting LLM Translation Skills without General Ability Loss via Rationale Distillation"
_ICLR.cc/2025/Conference — Submitted to ICLR 2025_

### Official Review · Reviewer_cw7y · 2024-10-27

**Soundness:** 2
**Presentation:** 3
**Contribution:** 2
**Rating:** 3
**Confidence:** 4

**Summary:**

This paper introduces Rationale Distillation (RaDis), a new approach for fine-tuning Large Language Models (LLMs) on machine translation (MT) tasks. Unlike traditional fine-tuning methods that rely solely on parallel corpora, RaDis uses LLM-generated rationales to enhance MT performance while maintaining the LLM’s general capabilities, such as instruction-following and reasoning.

**Strengths:**

RaDis is both straightforward and effective, leveraging the language model itself to generate explanations (rationales) for fine-tuning. Experimental results demonstrate that this method preserves the model’s general abilities while enhancing MT performance, although its MT performance remains slightly below established baselines like ALMA.

**Weaknesses:**

Firstly, the proposed method lacks novelty from my perspective, as leveraging LLMs to generate data from their own distribution for continual fine-tuning has been extensively explored in prior research (e.g., various PPO/DPO variants along with works on chain-of-thoughts that prompt LLM to generate reasoning trace, etc.,). Moreover, based on my experiences with machine translation (MT) and reinforcement learning with human feedback (RLHF), the generative quality of a 7B model is often unreliable, typically requiring filtering processes (i.e., from a reward model). Incorporating all rationales generated by the LLM without any scoring, filtering, or grounding is likely to reinforce the model's own hallucinations. Even if this approach preserves the model’s instruction-following abilities, the practical utility of such fine-tuning remains questionable.

Secondly, the rationales derived from a translation dataset may lack diversity. A comparison or discussion with existing work, such as TOWER [1], is necessary. TOWER achieves high MT performance by utilizing parallel corpus and diverse instruction-following datasets during fine-tuning, which appears more effective on benchmarks (and the idea is more straightforward than rationales generation). Therefore, I believe another baseline that simply uses parallel data for fine-tuning + diverse instruction-following dataset for preserving the model's pretrained ability is needed.

[1] Alves et al., (2024). TOWER: An Open Multilingual Large Language Model for Translation-Related Tasks

**Questions:**

1. some comparison and discussion on previous work like TOWER will be useful.
2. I am curious if there is any analysis of the generated rationales’ quality. I see in Table 4 there are some categorization but are all the generated responses relevant? I doubt that a 7B model can give very high-quality rationales for all translation pairs in the training data.
3. I like the analysis in section 5.2 and it actually resonates with my points in weakness. Fine-tuning model on its self-generated response is most helpful to preserve its instruction-following ability. However, I believe simply using all self-generated rationales is also limited. I am wondering if any on-policy-based mechanism for rationales generation is tried.
4. Overall, I find the experiments comprehensive but the improvement is not very surprising. The experiment also lacks an obvious baseline: using a combination of parallel corpus for MT and instruction-following dataset for preserving general ability (like the approach used in TOWER)

---

> ### Author Response · Authors · 2024-11-19
> **Response to Reviewer cw7y**
>
> Thanks for your efforts to provide insightful comments. We address your concerns point by point below.
>
> > W1a: The novelty of our work.
>
> We would like to clarify that RaDis is a **Continual Instruction Tuning (CIT)** method and we mainly ground it to translation tasks in our paper. We believe that our paper made several unique contributions and its novelty is also recognized by the other three Reviewers:
>
> First, as demonstrated in General Response 1, RaDis introduces a novel paradigm for building LLMs with both translation proficiency and general abilities. Specifically, previous works (e.g., BayLing, TowerInstruct) have primarily adopted a **multi-task training paradigm**, which fine-tunes a base LLM on both translation and general instruction data together. In contrast, RaDis follows a **continual learning (CL) paradigm**, fine-tuning chat LLMs on translation data while leveraging continual learning techniques to mitigate the forgetting of general abilities. Our experiments show the superiority of this paradigm in preserving general abilities, while also demonstrating its competitive performance and future potential in translation tasks.
>
> Second, RaDis is a novel CIT method with a strong motivation. To the best of our knowledge, we are the first to observe that LLMs tend to generate detailed rationales for translation tasks without any explicit chain-of-thought (COT) prompting. Furthermore, we are also the first to leverage this observation by proposing a CIT method that distills self-generated rationales.
>
> Regarding the specific concern you raised, RaDis is a continual learning (CL) method and therefore is not related to prior RLHF works. The only similarity between the two is that both approaches use self-synthesized data. However, the use of the model itself to generate training data is a widely adopted approach in the ML/NLP field, especially in distillation-related methods. Given that the core contribution of RaDis lies in the innovative construction of distillation data, we do not believe this diminishes the novelty of our approach.
>
> In terms of COT distillation methods in the reasoning field, we have included a dedicated section in the Related Work to contextualize our approach. Specifically, prior research in the COT reasoning domain typically involves prompting an external model to generate a reasoning path, which is then used to train the student model to enhance its reasoning capabilities. In contrast, RaDis leverages intrinsic, self-generated rationales as replay data in CL to preserve the model's original abilities. These distinctions make RaDis fundamentally different from prior studies and provide fresh insights into the application of CL for LLMs.
>
> > W1b&Q2&Q3: The quality of self-generated rationales. Using reward models for on-policy optimization.
>
> Your suggestion to use a reward model to evaluate the rationale quality is truly insightful. As shown in Table 5, higher-quality rationales can indeed lead to better translation performance. However, we hope to argue that the goal of distilling self-generated rationales is to preserve the original output distribution. From that perspective, the optimal choice is to utilize exactly the same output sampled from the model itself, which shall minimize the distribution gap. As shown in Table 5, Llama-3-70B, a stronger and larger model than Mistral-v0.2-7B, could generate higher-quality rationales. However, fine-tuning with these high-quality rationales does not lead to better general abilities. This is mainly because these rationales are out-of-distribution and would disrupt the output distribution of the student LLM, which has proven to cause forgetting [1].
>
> It is possible that sampling rationales from the output distribution and choosing high-quality ones for training may not cause a huge disruption to the model's output distribution. However, such a reward model needs to be trained for each backbone LLM and requires substantial human annotation. Unfortunately, we do not have the resources to do this at the moment. We leave this investigation to our future work.
>
> > W2&Q1&Q4: Comparisons to Tower and multi-task training baseline.
>
> Please refer to General Response 1, where we demonstrate that RaDis outperforms both Tower and multi-task training baseline in general abilities and performs competitively in translation performance.
>
> [1] Yang, Zhaorui, et al. "Self-distillation bridges distribution gap in language model fine-tuning." Proceedings of the 62nd Annual Meeting of the Association for Computational Linguistics (Volume 1: Long Papers). (2024).

---

> > ### Author Response · Authors · 2024-11-22
> > **Response to Reviewer cw7y (Cont.)**
> >
> > > Traing with self-generated rationales may reinforce the model's own hallucinations.
> >
> > Regarding your specific concern about hallucinations, we conducted experiments using Factool [2], a task- and domain-agnostic framework designed to detect factual errors in texts generated by large language models. We evaluated the models' factuality on two benchmarks: knowledge-based QA and HumanEval.
> >
> > Specifically, for knowledge-based QA, Factool employs GPT-4 to extract factual claims from the model’s responses and uses Google Search to verify the factual accuracy of these claims. This methodology ensures a robust and comprehensive assessment of the model’s factuality.
> >
> >
> > Table c1: results for model factuality.
> > |                       | KBQA | HumanEval |
> > |-----------------------|------------------------|-----------|
> > | Mistral-v0.2-Instruct | 64.55                  | 36.59     |
> > | RaDis                 | 66.08                  | 35.97     |
> >
> > As shown in `Table c1`, RaDis maintains claim-level factuality on knowledge-based QA and preserves the Pass@1 rate on HumanEval. We believe these results demonstrate that RaDis does not exacerbate hallucination issues, addressing your concern effectively.
> >
> > [2] Chern, I., et al. "FacTool: Factuality Detection in Generative AI--A Tool Augmented Framework for Multi-Task and Multi-Domain Scenarios." arXiv preprint arXiv:2307.13528 (2023).

---

> > > ### Comment · Reviewer_cw7y · 2024-11-25
> > > **Official Comment by Reviewer cw7y**
> > >
> > > I want to thank reviewer for providing additional experiments. I would like to keep my score because of the following reasons:
> > >
> > > > CIT v.s. Multitask training:
> > >
> > > I think you use the term continual learning because you re-use instruction-tuned model (mistral/qwen2 chat). The comparison is still problematic because ideally you should compare
> > > - (a) base model (mistral/llama/qwen base) + MT finetuning
> > > - (b) base model + multitask finetuning with MT and other instruction-tuning data
> > > - (c) (stage 1: finetune base model on other instruction-tuning data) + (stage 2: RaDis finetuning)
> > >
> > > When you make comparisons, you reuse previous work for (a) and (b) (i.e. using ALMA, Tower, Bayling etc.,), but when constructing (c), you use a more capable instruction-funed model. Therefore, you improvement on general instruction-following benchmark is not fair to compare with previous baselines as it largely leverage the instruction-tuning from a more advanced model.
> > >
> > > More importantly, multitask finetuning should be more scalable and general (as translation is essentially just one task), therefore I am not very excited by progress in continual learning (this is my personal opinion that I do not factor into my rating)
> > >
> > > > RaDis is a continual learning (CL) method and therefore is not related to prior RLHF works
> > >
> > > Your method might not be relevant to RLHF but it is definitely one kind of post-training method for LLM. As you mentioned, the use of the model itself to generate training data is a widely adopted approach in the ML/NLP field. And currently, RLHF-relevant study achieves impressive post-training results. That is why I think simply prompting and utlizing all model's self-generated content is no longer a promising direction.
> > >
> > > You mentioned "the optimal choice is to utilize exactly the same output sampled from the model itself, which shall minimize the distribution gap". The distribution mismatch is a common problem and my point is that you should (1) use model's self-generated rational and sample multiple explanations for filtering (2) use another model to filter/reweight the generation. Directly generating explanations with another model will surely result in distribution mismatch (and this is already studied in on-policy and off-policy RL).
> > >
> > > > Conclusion
> > >
> > > I still really appreciate the additional experiments conducted by the authors. I think I don't want to push for acceptance mainly because I don't enough new insights compared to current post-training methods. The use of self-generated data to preserve a well-trained chat-LLM's instruction following ability is in no way surprising to me. Moreover, compared to the current methods for post-training (SFT with multitask data + aligning to human feedback + inference-time reasoning boost), the proposed method attempts to mix the SFT and alignment stage through RaDis by simply prompting model to generate some explanations. I don't think this direction is promising because it is a simpler version of current (SFT+alignment) post training framework where the sampling process is unconstrained.

---

> ### Author Response · Authors · 2024-11-29
>
> Thank you for your response. We would appreciate the opportunity to engage in a more detailed discussion with you.
>
> > Fairness regarding utilizing chat LLMs.
>
> The traditional training approach is **unable** to leverage chat LLMs. Because they suffer from severe forgetting caused by continual pre-training (CPT) or vanilla fine-tuning. In contrast, RaDis allows for the retention of the general abilities of chat LLMs. We believe that utilizing accessible chat LLMs represents a valuable contribution, as it addresses the limitations of previous methods.
>
> > Utilizing an RM to filter rationales in RaDis.
>
> We acknowledge that implementing an 'RLHF-based' version of RaDis could potentially improve results. However, the substantial resources required to train an RM were not available to us. Besides, introducing an RM will also raise questions on whether the general abilities are preserved by rationales or learned from the RM.
>
> Additionally, the specific implementation is not the focus of this paper. Our key contribution is introducing **rationale** in CL, which brings new insights on relying on the **reasoning ability** of LLMs to **connect learned and new knowledge** and alleviate forgetting. The filtering technique is **orthogonal** to our contribution (the rationales are still self-generated whether they are filtered or not) and does not hurt novelty.
>
> > Rationale quality.
>
> To further address your concern regarding the rationale quality, we randomly sampled 200 sentence pairs and examined their corresponding rationales. Our analysis found that most of the rationales are highly related to the corresponding sentence pairs.This is because the model can refer to the reference translation when generating rationales, which is consistent with prior work [1] showing that post-rationalization helps reduce hallucinations. We have uploaded the rationale samples as supplementary materials for further reference.
>
> > Whether RaDis is just a simplified version of current aligning approach.
>
> We would like to emphasize the fundamental differences between RaDis and RLHF.
>
> The objective of RaDis is to **fine-tune the model for a specific task while preserving its original abilities**. This involves training with task data. Self-generated data is used to **mitigate forgetting**.
>
> In contrast, RLHF aims to **aligning model behavior with human preferences**, with little or no task-specific annotated data. The model learns from **human feedbacks**. Self-generated data is used as a **bridge to convey human preferences**.
>
> While both approaches use self-generated data, it **serves different purposes, in different ways, with distinct motivations**. Therefore, we respectfully disagree with the reviewer’s characterization of RaDis as a simplified version of current aligning approach. As using synthetic data for training is currently one of the most common methods for fine-tuning LLMs, it would not be fair to claim that a paper lacks novelty because it utilizes synthetic data for training. While the training approach may seem similar on the surface, we believe that the reviewer may have overlooked the novel insights introduced by RaDis (**utilizing rationales to alleviate forgetting**), as well as the differences between our method and the existing post-training framework.
>
> > Why RaDis presents non-trival contribution
>
> We believe the reviewer's claim that 'using self-generated data to preserve a chat-LLM's ability is not surprising' oversimplifies the problem and our approach.
>
> First, RaDis involves **SFT with task data**. In contrast, in RLHF (on-policy), the model is fine-tuned exclusively with self-generated data, with 10-100x smaller learning rate. Given these difference, the findings that forgetting can be easily mitigated with self-generated data in the RLHF phase **do not necessarily generalize to the SFT phase**.
>
> Second, **not all self-generated data is equally effective in alleviating forgetting**. As demonstrated in our experiments, SDFT also uses self-generated data during training. However, its effectiveness in mitigating forgetting is clearly inferior to that of RaDis.
>
> Finally, **self-generated data may hinder new task learning**. As demonstrated in our experiments, Seq-KD and SDFT—two baselines that also utilize self-generated data—failed to improve the model's translation performance, while RaDis **balances both learning and consolidating**.
>
> The clarifications above demonstrate that achieving effective CIT, which requires both alleviating forgetting and learning new knowledge, with synthetic data is not a trivial problem. Therefore, we sincerely hope the reviewer could reconsider the novelty of our proposed approach. If the reviewer still has concerns, we would sincerely appreciate references to prior works that may address similar ideas.
>
> [1] Chen, Xiao, et al. "Post-Semantic-Thinking: A Robust Strategy to Distill Reasoning Capacity from Large Language Models." arXiv preprint arXiv:2404.09170 (2024).

---

> ### Author Response · Authors · 2024-12-03
> **Kind reminder**
>
> Dear Reviewer cw7y,
>
> Thank you for the time and effort you've dedicated to reviewing our work. As the discussion phase comes to a close, we kindly ask that you review our follow-up responses (including the new general responses), which aim to address your latest questions.
>
> **While we recognize there may still be differences in opinion, we deeply appreciate the thoroughness of your feedback. We understand that providing multiple rounds of detailed commentary is both rare and invaluable. Our responses aim to foster a deeper discussion of differing viewpoints. Regardless of whether you choose to maintain or revise your score, we fully respect your decision.**
>
> Thank you again for your time and consideration.
>
> Best regards,
>
> Authors

---

### Official Review · Reviewer_M6wr · 2024-10-28

**Soundness:** 2
**Presentation:** 3
**Contribution:** 3
**Rating:** 6
**Confidence:** 4

**Summary:**

This paper proposes a novel method, RaDis, which uses self-generated rationales for sequence-level distillation. It can preserve the model's capabilities in general domains and safety while fine-tuning downstream tasks, addressing the problem of catastrophic forgetting. The paper conducts machine translation experiments on two 7B-sized models to illustrate the effectiveness of the method, and the analysis section provides the detailed observation of the experimental results.

**Strengths:**

1. This paper proposes a novel method, RaDis, which has a clear and straightforward motivation, and the writing is easy to follow.
2. The experimental results demonstrate that this method is effective and significantly surpasses the related baselines.

**Weaknesses:**

1. This method effectively maintains the generalization ability and safety of the models during downstream task fine-tuning. However, most translation performance degrades compared to the original fine-tuning baseline, which will limit its applicability in real-world scenarios. Additionally, the comparison of translation performance between the backbone and RaDis in lines 81-83 is unfair and the comparison with the vanilla fine-tuned model is more reasonable and informative.
2. Considering the baseline distillation methods (SeqKD and SDFT) largely relies on the generation quality of the teacher model, it is necessary to conduct the experiments with teachers with better capacity, especially for SeqKD, which is rarely used in self-distillation manner.
3. In the analysis section, the explanations on the performance of RaDis with three different teachers needs a more detailed analysis, similar to that in Table 4. Particularly, it is necessary to clarify why the machine translation performance and the general performance yield contrasting conclusions (e.g., the self-generated setting achieve best general performance and the worse translation performance among the three settings).
4. Typo: line 507, “SDFT greatly enhances” -> “RaDis greatly enhances”

**Questions:**

1. In the context of machine translation, the emphasis on faithfulness over diversity suggests that paraphrasing references may not efficently enhance translation performance, which aligns with the primary experimental results presented in Tables 1 and 2. Consequently, to facilitate a more equitable comparison between RaDis and SDFT, as well as to explore the potential of RaDis in other tasks, it is essential to evaluate their corresponding performance on logical reasoning tasks, including OpenFunctions, GSM8K, and HumanEval.
2. See the weeknesses.

---

> ### Author Response · Authors · 2024-11-19
> **Response to Reviewer M6wr**
>
> We appreciate your questions and value your feedback. Below, we provide detailed explanations to address your concerns.
>
> > W1a: Applicability in real-world scenarios.
>
> First, we would like to clarify that RaDis is a continual learning (CL) method. A key challenge in CL is known as the **stability-plasticity trade-off**, where excessive
> learning plasticity or memory stability can compromise each other. As a result, achieving the same performance as vanilla fine-tuning while preserving all of the model's original abilities is inherently difficult [1,2].
>
> Table m1: Experimental results with Qwen2.5-Instruct.
>
> |                    | X-EN  | EN-X  | Alpaca Eval | Alpaca Eval 2 | AdvBench | GSM8K |
> |--------------------|-------|-------|-------------|---------------|----------|-------|
> | Qwen2.5-Instruct           | 80.90 | 80.50 | 88.46       | 31.55         | 99.81    | 87.72 |
> | +Vanilla-FT        | 82.21 | 82.95 | 4.87        | 2.95          | 27.12    | 26.61 |
> | +RaDis             | 82.13 | 82.81 | 85.62       | 27.91         | 99.04    | 88.78 |
>
> In our original paper, the average translation performance with Mistral-v0.2 as the backbone LLM is 84.31/82.33 for Vanilla-FT and 84.39/81.94 for RaDis. The average gap between RaDis and Vanilla-FT is only **0.155 COMET22** across eight translation directions. As shown in `Table m1`, switching the backbone to Qwen2.5-Instruct narrows this gap further to **0.11 COMET22**, which is a relatively small performance difference. On the other hand, RaDis consistently outperforms Vanilla-FT in preserving general abilities, retaining 82.31% more of these abilities on average. Given these results, we believe that RaDis is competitive in task-specific performance and superior in general task performance, making it a highly effective fine-tuning method applicable to real-world scenarios.
>
> > W1b: Comparison between RaDis and the backbone LLM.
>
> Thank you for your suggestion. In our original paper, we compared RaDis with the backbone LLM primarily to demonstrate that it enhances translation performance while preserving general capabilities. Comparing RaDis with Vanilla-FT highlights that RaDis not only achieves comparable translation performance but also significantly mitigates the forgetting of general abilities. We believe that both comparisons convey the same core message: RaDis is an effective method for continual instruction tuning. Following your suggestion, we have modified the sentence to improve clarity.
>
> > W2: Utilizing a stronger teacher model.
>
> We are happy to address the concern regarding the use of a stronger teacher model. As explained in Section 4.2, Seq-KD and SDFT are part of the Continual Instruction Tuning (CIT) baseline. In the continual learning setting, only the dataset for the current task and the model to be trained are available. As we focus on preserving the ability of the **original LLM** instead of learning from a stronger teacher, most CL approaches that involve knowledge distillation are limited to self-distillation [1]. Therefore, we believe our experimental setup is well justified.
>
>
> [1] Wang, Liyuan, et al. "A comprehensive survey of continual learning: theory, method and application." IEEE Transactions on Pattern Analysis and Machine Intelligence (2024).
>
> [2] Wu, Tongtong, et al. "Continual learning for large language models: A survey." arXiv preprint arXiv:2402.01364 (2024).

---

> ### Author Response · Authors · 2024-11-19
> **Response to Reviewer M6wr (Cont)**
>
> > W3: Clarification on the ablation results
>
> Here we provide further clarification on the results presented in Section 5.2. There appear to be some typos in Lines 462-463: "These results suggest that models can learn stronger ~~general~~ （translation） capabilities through higher-quality rationales, which leads to better ~~overall~~ performance." Our finding is that translation performance can benefit from higher-quality rationales generated by stronger teacher models, which aligns with previous work on chain-of-thought distillation methods in reasoning fields [4].
>
> Regarding the preservation of general abilities, we found that using the model itself as the teacher is the most effective approach. This is primarily because rationales generated by an external teacher model tend to be out-of-distribution relative to the student model. Fine-tuning with these out-of-distribution rationales can significantly corrupt the model’s output distribution, which has been shown to lead to forgetting [3].
>
> > Q1: Testing RaDis on broader tasks and comparing it to SDFT.
>
> In our experiments, we observed that paraphrasing translation references generates completely irrelevant sentences when using Mistral-v0.2 as the backbone, which leads to significantly worse translation results.
>
> In General Response 2, we conducted experiments on code generation tasks using the Magicoder dataset. Compared to translation, code generation is more deterministic, and our results show that RaDis still outperforms SDFT in this task. Please refer to General Response 2 for more detailed results and analysis.
>
> > Typos.
>
> Thank you for pointing this out. We will correct these typos in a revised version of our paper.
>
> [3] Yang, Zhaorui, et al. "Self-distillation bridges distribution gap in language model fine-tuning." Proceedings of the 62nd Annual Meeting of the Association for Computational Linguistics (Volume 1: Long Papers). (2024).
>
> [4] Wang, Peifeng, et al. "SCOTT: Self-Consistent Chain-of-Thought Distillation." Proceedings of the 61st Annual Meeting of the Association for Computational Linguistics (Volume 1: Long Papers). 2023.

---

> ### Author Response · Authors · 2024-11-30
> **Looking forward to your feedback!**
>
> Dear Reviewer M6wr,
>
> We appreciate your insightful feedback. Following your suggestions, we have carried out further experiments and made revisions to the paper. As we are approaching the end of the discussion phase, we would like to know if our responses have resolved your concerns. We look forward to your response.
>
> Best, Authors

---

> > ### Comment · Reviewer_M6wr · 2024-12-02
> >
> > Thank you for your response. Most of my concerns have been addressed and I have changed my score to reflect this.

---

> > > ### Author Response · Authors · 2024-12-02
> > > **Thank you for raising the score!**
> > >
> > > Dear Reviewer M6wr,
> > >
> > > Thank you for raising the score! Once again, we sincerely appreciate the valuable feedback you provided, which has been incredibly helpful in improving our work.
> > >
> > > Best,
> > >
> > > Authors

---

### Official Review · Reviewer_JdnZ · 2024-11-02

**Soundness:** 4
**Presentation:** 4
**Contribution:** 3
**Rating:** 6
**Confidence:** 4

**Summary:**

The paper introduces RaDis (Rationale Distillation) -- a self-distillation technique to help instruction-tuned language models learn new tasks without losing their general capabilities or compromising safety alignment. The method first generates rationales - explanations for the instruction-response pairs of the target task using the same language model and these rationales serve as replay data that help retain the original capabilities by fairly capturing the data distribution. The generated rationales and instruction-response pairs are then used for subsequent training of the same language model on the new task. The authors conduct experiments for the task of machine translation and demonstrate that it improves the translation proficiency of models while preserving its overall performance on other tasks.

**Strengths:**

1. The authors propose a straightforward method for LLMs to learn the translation task (in this study) without losing prior abilities by combining standard fine-tuning on reference translations with self-distillation on generated rationales.

2. The experimental results demonstrate translation improvements across 4 language pairs on 2 different models.

3. The authors study various aspects of the proposed method to understand the influence of these aspects on downstream and overall performance.

**Weaknesses:**

1. The paper only focuses on improving translation proficiency in LLMs without losing existing capabilities. However, I think the proposed method is more general and can work for other tasks. It should also be experimented on other tasks (e.g. summarization, open-ended QA, MQM annotation, etc) for a more comprehensive study.

2. In section 5.2, the authors compare the Mistral 7B model (student) with rationales generated from different models. However, this comparison may not be fair and it might be more appropriate to compare the rationale generation from a bigger model within the same family. For instance, the Llama 3 7B model (student) with rationales generated from Llama 3 70B. Could you clarify the reasons for not using models from the same families (e.g. Llama 3)?

**Questions:**

The proposed method generates a rationale for each sample in the training set. Did you experiment with ablating whether you need a rationale for each sample to be included in the training set? How well does the proposed method perform when rationales aren’t available for every sample in the training set? What is the tradeoff between having rationales for all samples versus just a subset of *k* samples (e.g., 10%, 20%, 50% and 100% samples with rationales)?

---

> ### Author Response · Authors · 2024-11-19
> **Response to Reviewer JdnZ**
>
> Thanks for your recognition of our work and insightful questions. We believe they hold significant value for our work. We hope the following responses can help address your concern.
>
> > W1: Testing RaDis on broader tasks.
>
> Please refer to General Response 2, where we conducted experiments on code generation tasks using the Magicoder dataset. The results demonstrate the superiority of RaDis and highlight its potential.
>
> Given that our current paper primarily focuses on translation tasks, and considering the limited time during the discussion phase, testing RaDis on a broader range of tasks may not be feasible at this stage. However, we greatly appreciate your insightful suggestion and plan to further develop RaDis as a robust, general-purpose continual instruction tuning method in our future work.
>
> > W2: Experiment setting in Section 5.2.
>
> We are happy to clarify the rationale behind the specific setting in Section 5.2. The primary goal of this section is to conduct an ablation study to analyze which aspects of RaDis contribute most to its success. As noted in the paper, we focus on two key factors: rationale quality and the self-distillation property. However, since these two factors are generally intertwined in the generated rationales, we needed to design specific experiments to separate them. To do so, we incorporated a stronger teacher from a different model family.
>
> Table j1: Rationale for selecting different teachers.
> |     Teacher Model        |     Rationale Quality |     Self-distillation |
> |--------------------------|-----------------------|-----------------------|
> |     Self                 |     Moderate          |     Yes               |
> |     LLaMA-2-Chat-7B      |     Moderate          |     No                |
> |     LLaMA-3-70B-Instruct |     High              |     No                |
>
> As shown in `Table j1`, with Mistral-v0.2 as the backbone model, we can categorize the teacher models along two axes: rationale quality and whether they ensure the self-distillation property. It is straightforward to observe that a larger and more capable teacher model tends to generate higher-quality rationales. However, ensuring that the teacher model does not satisfy the self-distillation property requires more careful design. Since models from the same family often share a significant portion of their training data, we opted to use models from completely different model families. This ensures that the generated rationales are out-of-distribution relative to the student model. By doing so, we can effectively separate the rationale quality and self-distillation property and access their contributions, respectively.

---

> ### Author Response · Authors · 2024-11-22
> **Response to Reviewer JdnZ (Cont.)**
>
> > Q1: Ablation experiments on the percentage of examples with rationales
>
> Table j2: Ablation experiments on the percentage of examples with rationale.
> |                       | X-EN      | EN-X      | Alpaca Eval | Alpaca Eval 2 | AdvBench  | GSM8K    |
> |-----------------------|-----------|-----------|-------------|---------------|-----------|----------|
> | Mistral-v0.2-Instruct | 80.84     | 67.79     | 84.91       | 15.09         | 68.46     | 41.62    |
> | 0%                    | **82.33** | 84.31     | 6.07        | 1.02          | 4.23      | 0.23     |
> | 25%                   | 82.25     | 84.21     | 79.22       | 10.84         | 62.69     | 34.5     |
> | 50%                   | 82.20     | 84.51     | 79.21       | 9.62          | **63.65** | 35.1     |
> | 75%                   | 82.03     | 84.35     | 79.3        | 10.43         | 63.54     | 35.41    |
> | 100%                  | 81.94     | **84.39** | **80.34**   | **11.05**     | 62.12     | **41.7** |
>
> Your suggestion to include an ablation study on rationales has been constructive. We conducted experiments with varying percentages of examples containing rationales (25%, 50%, 75%) and would like to share the results.
>
> First, the results show that RaDis achieves the best overall performance on general tasks when rationales are provided for the entire training dataset. However, adding rationales to just 25% of the training data significantly mitigates forgetting on general tasks. Interestingly, the effectiveness of rationales appears to vary across task types: instruction-following and safety tasks are the easiest to preserve, while math reasoning tasks are the most challenging.
>
> Moreover, scaling the percentage of data with rationales from 25% to 75% does not result in a linear improvement. Notably, the model's performance changes dramatically in two key intervals: from 0% to 25% and from 75% to 100%. These findings highlight that the inclusion of rationales has a non-linear impact on performance, with significant gains observed at the extremes.
>
> We believe this phenomenon could be explained as follows:
>
> 1. **Superficial Alignment Hypothesis**: As defined in LIMA [1], the Superficial Alignment Hypothesis suggests that most of a language model’s abilities and knowledge are acquired during pre-training, while post-training primarily focuses on refining the model's style and format. Building on this hypothesis, we interpret our ablation results to indicate that the forgetting of superficial alignment can also be mitigated superficially. For instance, incorporating a small number of rationales (25%) is sufficient to recover the model’s performance on instruction-following and safety tasks. However, this level of rationale integration is adequate for more complex skills, such as math reasoning, which does not obey the Superficial Alignment Hypothesis.
>
> 2. **Distribution Gap of Training Data**: Traditionally, when training with mixed data from two tasks, the results are expected to be a linear combination of training each task independently. However, due to the vast number of parameters in LLMs, these models can effectively memorize different tasks in separate parameter spaces based on their input patterns. This phenomenon has also been observed in the Llama3 technical report, where researchers from Meta utilized distinct system prompts to toggle the model between different modes (e.g., text or speech).
>
>     As a result, the mixup ratio of training data may not be a critical parameter. Instead, the primary factors are the tasks included in the training set and whether they are in-domain or out-of-domain. When 0% of the data contains rationales, the model is trained solely on translation tasks (out-of-domain), leading to severe forgetting of general abilities. By incorporating a mix of translation and rationale-augmented translation (which can be considered a form of instruction-following translation task), the model learns both tasks simultaneously. However, the existence of an out-of-distribution task still leads to forgetting. When the entire training set includes rationales, the model is trained predominantly on in-domain tasks, significantly alleviating the forgetting of general abilities.
>
> We hope the experimental results and analysis presented above adequately address your question. If you have any further concerns, we would be delighted to engage in a more in-depth discussion with you.
>
> [1] Zhou, Chunting, et al. "Lima: Less is more for alignment." Advances in Neural Information Processing Systems 36 (2024).

---

> > ### Comment · Reviewer_JdnZ · 2024-11-26
> >
> > I would like to thank the authors for their detailed response to my concerns. The response indeed brought interesting findings about the trade-off between the proportion of rationales necessary to achieve reasonable performance. Most of my concerns have been addressed effectively. However, I believe it would be valuable to broaden the scope of the study by including experiments across additional task setups. While the authors report experimental results for code generation, testing the proposed method on a wider range of tasks would be extremely valuable and comprehensive. I would like to maintain my original rating.
> >
> > This is not a criticism, but I agree with Reviewer pUWZ that the authors should not strictly view continued fine-tuning and multi-task learning as separate paradigms, as there are some similarities. The assumption of access to a stream of single-task data at a time may not be entirely practical or realistic. The authors should also think about how to extend the proposed approach to a more general setup similar to multi-task learning, where there are no task-specific distinctions among examples and the order of examples from different tasks is treated as arbitrary and non-sequential.

---

> ### Author Response · Authors · 2024-11-30
> **Thank you for your response!**
>
> Thank you for your response! We're glad to hear that many of your issues have been resolved. We are conducting experiments on additional tasks and will provide the results as soon as possible. Besides, we've provided a new general response, explaining why we would view CL and MTL two different learning paradigm and why investigating CL for LLM is crucial and irreplaceable in the application of LLMs.

---

> ### Author Response · Authors · 2024-12-02
> **Results of new experiments.**
>
> Thank you once again for your feedback! We have conducted additional experiments on TriviaQA (open-ended QA) and OpenFunctions (function call). Below are the results from these experiments:
>
> |                 | **OpenFunctions** | **Alpaca Eval** | **Alpaca Eval 2** | **AdvBench** | **gsm8k** |
> |-----------------|-------------------|-----------------|-------------------|--------------|-----------|
> | **Llama-2**     | 16.07             | 71.4            | 9.66              | 100          | 24.34     |
> | **+Vanilla-FT** | 29.46             | 36.52           | 3.41              | 100          | 20.69     |
> | **+SDFT**       | 32.14             | 68.51           | 7.82              | 100          | 21.00        |
> | **+RaDis**      | **33.04**         | **69.52**       | **8.79**          | 100          | **24.56** |
>
> |                  | **TriviaQA** | **Alpaca Eval** | **Alpaca Eval 2** | **AdvBench** | **gsm8k** |
> |------------------|--------------|-----------------|-------------------|--------------|-----------|
> | **Mistral-v0.2** | 53.4         | 84.91           | 15.09             | 68.46        | 41.62     |
> | **+Vanilla-FT**  | 57.9         | 29.82           | 3.09              | 30.38        | 9.86      |
> | **+SDFT**        | 53.83        | 79.47           | 10.4              | 36.35        | **40.64** |
> | **+RaDis**       | **58.71**    | **83.95**       | **14.22**         | **64.81**    | 39.27     |
>
> As demonstrated in the tables above, RaDis effectively alleviates forgetting and outperforms SDFT on both tasks. Additionally, it achieves superior performance on the fine-tuned task compared to Vanilla-FT. These results support our claim that rationales play a crucial role in bridging learned and new knowledge, thereby enhancing both task learning and knowledge retention.

---

> ### Author Response · Authors · 2024-12-03
> **Kind Reminder**
>
> Dear Reviewer JdnZ,
>
> Thank you again for the time and effort you’ve dedicated to reviewing our work. As the discussion phase is coming to a close, we kindly ask if you could review our follow-up responses. **We’re grateful that you’ve mentioned most of your initial concerns have been addressed by prior rebuttal, and we have also provided replies to your follow-up requests for testing our approaches with broader tasks.**
>
> **If there are no further concerns, we would be grateful if you could reconsider the score.**
>
> Thank you for your time.
>
> Best regards,
>
> Authors

---

> > ### Comment · Reviewer_JdnZ · 2024-12-03
> >
> > Thank you for conducting experiments on new tasks and addressing all of my concerns. I have updated the score.

---

> > > ### Author Response · Authors · 2024-12-03
> > >
> > > Thank you for your quick response and positive feedback. :)
> > >
> > > We’re glad to hear that our response addressed your concerns. However, we’ve noticed that the rating has not been updated. We would truly appreciate your support in this matter and wanted to kindly inquire if there are any remaining issues or feedback we can address to further clarify or improve the manuscript. Your insights are invaluable to us, and we’re more than happy to provide any additional information or revisions as needed.
> > >
> > > Thank you again, and we look forward to your response. Wishing you a pleasant day!

---

> > > > ### Comment · Reviewer_JdnZ · 2024-12-03
> > > >
> > > > I have updated the soundness rating but prefer to maintain the overall rating.

---

> > > > > ### Author Response · Authors · 2024-12-03
> > > > > **Thank you**
> > > > >
> > > > > Dear Reviewer JdnZ,
> > > > >
> > > > > Thank you for updating the soundness rating and for your continued engagement. We fully respect your decision to maintain the overall rating. We appreciate the time and effort you've dedicated to reviewing our manuscript and are grateful for your thoughtful feedback.
> > > > >
> > > > > Best regards,
> > > > >
> > > > > Authors.

---

### Official Review · Reviewer_pUWZ · 2024-11-05

**Soundness:** 3
**Presentation:** 3
**Contribution:** 2
**Rating:** 5
**Confidence:** 5

**Summary:**

The paper proposed rationale distillation, with a purpose on improving LLM-based models translation performance while keeping the general instruction-following abilities. The method first generates the explanation of the translation and use the synthetic data for fine-tuning along with the translation pairs.

**Strengths:**

The paper successfully did what they proposed, improving the translation performance with the proposed method can significantly reduce the damage in general instruction-following tasks.

**Weaknesses:**

1. The paper's goal is good —it aims to generalize translation models based on LLMs to other domains such as conversational tasks. However, the attempt to balance both translation quality and generalization does not yield a model that is truly useful, placing the paper in an awkward position. For example, RaDis performs substantially worse than its backbone model on conversations and instruction following, and its translation performance is inferior to that of translation-specific models. While I acknowledge that comparing RaDis with ALMA in terms of translation is not entirely fair due to differences in models and training data, this raises another issue: the model's performance is highly dependent on the backbone model. Consequently, RaDis cannot be effectively applied to low-resource languages like Icelandic, which were excluded from the training data. Achieving effective machine translation in LLMs requires substantial effort in multilingual pretraining and alignment; without this, English-centric LLM translation performance is less interesting. Despite the paper's claims of keeping both translation high performance and retaining generality, the translation quality is not top due to the high dependency of the backbone model, and the performance on other tasks is also worse.

2. Inference speed: Regarding inference speed, if the model is trained to output both a reference and an explanation, does this mean the generated output always includes both, requiring the user to extract the translation from the combined output? If so, this could significantly reduce inference speed. It would be beneficial to see a comparison of inference speeds and an analysis of how many additional tokens the model generates compared to translation-specific LLMs.

3. Missing baselines: A fundamental baseline would involve training on a combined dataset of translation parallel data and instruction-following data. Using synthesized data may be unnecessary given the abundance of existing data that can be combined with parallel data. However, I did not find results corresponding to this baseline in the paper. Including such comparisons would strengthen the evaluation and provide a clearer understanding of the model's performance relative to more straightforward approaches.

**Questions:**

N/A

---

> ### Author Response · Authors · 2024-11-19
> **Response to Reviewer pUWZ**
>
> Thanks for your insightful questions and they hold significant value for our work. We appreciate your recognition of the meaning of our research topic—building LLMs that excel in both translation proficiency and general ability. We hope the following discussion will address your concern and show the advantages and potential of our approach.
>
> > W1a: The performance seems to be worse on both translation and general ability.
>
> We would like to argue that this comparison is not entirely fair, as it is comparing RaDis with two models in their respective advantage zones. As we have shown in the paper and the general response. RaDis significantly outperforms LLM-based MT models like TowerInstruct in general abilities with 100x less training budget, and it also outperforms its original backbone in translation. Taken together, RaDis is the best model among those tested, successfully achieving both high translation proficiency and robust general abilities.
>
> Given the focus of our paper on building LLMs that excel in both translation proficiency and general ability, we believe it is more comprehensive to consider both aspects in the evaluation. As we have shown, **RaDis is indeed the best model from this perspective, achieving strong performance in both areas**.
>
> > W1b: The dependency on backbone LLM limits the multilingual translation performance.
>
> Regarding your concern that the translation performance of RaDis is limited by the choice of backbone LLM, we believe this limitation can be effectively addressed. As demonstrated in General Response 1, switching the backbone from Mistral to Qwen2.5 leads to substantial improvements in translation performance, effectively narrowing the gap caused by multilingual training. Moreover, with the emergence of open-source multilingual LLMs (e.g., Aya, Llama-3.1, Qwen2.5), the impact of multilingual training may diminish over time. As stronger backbone LLMs continue to develop, RaDis's translation proficiency can be further enhanced, while its superiority in general tasks will remain intact.
>
> Overall, from the perspective of building an "almighty" LLM, RaDis introduces a novel continual learning paradigm that is not only competitive but, in many respects, superior to the traditional multi-task training approach. It demonstrates significant advantages in preserving general abilities, while also showing strong potential in translation tasks.

---

> > ### Author Response · Authors · 2024-11-19
> > **Response to Reviewer pUWZ (Cont)**
> >
> > > W2: Inference speed issue
> >
> > We are happy to address your concern regarding the decoding overhead introduced by rationales.
> >
> > First, we would like to suggest that generating translations with rationales can be beneficial to end users. A key strength of LLMs is their ability to interact smoothly with users, and these 'verbose' outputs can actually enhance this interactivity. In fields such as recommender systems [1] and reasoning [2], LLM-generated explanations are proven helpful and favored by users. Similarly, we believe that generating rationales alongside translations can help users better understand the translated sentence, fostering a more explainable and transparent generation process.
> >
> > Table p1: Statistics of decoding speed on WMT22 Cs-En test set. The backbone LLM is Mistral-v0.2.
> >
> > |             | #Sample | Decoding Time(s) | Decoding Speed(sample/s) | COMET |
> > |-------------|---------|------------------|--------------------------|-------|
> > | Vanilla-FT  | 1448    | 55               | 26.33                    | 83.51 |
> > | RaDis       | 1448    | 123              | 11.77                    | 82.41 |
> > | +translation-only   | 1448    | 40               | 36.20                    | 82.41 |
> > | +sys-prompt | 1448    | 72               | 20.11                    | 82.80 |
> >
> > Second, the training process of RaDis offers several ways to control the generation of rationales, which can be particularly useful when high decoding speed is required:
> >
> > 1. **Stop generation at a delimiter token**: The construction of RaDis's training data involves concatenating the reference translation and self-generated rationales, separated by a delimiter token (e.g., \n). By stopping the generation at this token, the model can produce translations without rationales, thus achieving faster decoding without compromising translation performance. As shown in `Table p1`, when only generating the translation, RaDis' decoding speed is even faster than Vanilla-FT.
> > 2. **Controling rationale length with system prompts**: Leveraging the strong instruction-following ability preserved from the backbone LLM, RaDis can use system prompts to regulate the generation of rationales. For instance, as shown in `Table p1`, adding a system prompt like "_You are a professional translator, output the translation without explanation._" effectively suppresses the model’s tendency to over-generate, leading to faster decoding without sacrificing translation quality.
> >
> > > W3: Comparisons to Tower and multi-task training baseline.
> >
> > Please refer to General Response 1, where we demonstrate that RaDis outperforms both Tower and multi-task training baseline in general abilities and performs competitively in translation performance.
> >
> > [1] Lubos, Sebastian, et al. "LLM-generated Explanations for Recommender Systems." Adjunct Proceedings of the 32nd ACM Conference on User Modeling, Adaptation and Personalization. 2024.
> >
> > [2] Krause, Stefanie, and Frieder Stolzenburg. "From Data to Commonsense Reasoning: The Use of Large Language Models for Explainable AI." arXiv preprint arXiv:2407.03778 (2024).

---

> > > ### Comment · Reviewer_pUWZ · 2024-11-20
> > >
> > > I thank the authors for their response.
> > >
> > > I completely understand that the comparison between RaDis and other models like ALMA is not entirely fair. However, the authors themselves make such comparisons in the introduction — the motivation of this paper is to build a model that excels in both translation and reasoning. Even if the comparison is not perfectly fair, it is necessary because this is the claim the authors are making. Imagine the community needs to choose a translation model. Would people prefer ALMA, or RaDis which performs worse in translation but offers better reasoning capabilities?
> > >
> > > Referring to the authors' claim that "RaDis is indeed the best model from this perspective, achieving strong performance in both areas," I reviewed the results again. In Table 1, RaDis achieves a COMET score of 7.29 based on LLaMA-2, which is nice. However, in Table 3, it performs worse than LLaMA-2 in AlpacaEval and AlpacaEval 2.0. In Tables g1-2, while RaDis shows strong reasoning results with other backbone models, its translation performance is inferior to Tower （I understand it is not entirely fair). This is what my initial comment said that RaDis occupies an "awkward position."
> > >
> > > Secondly, the General Response 1 does not answer my question about low-resource language performance across other backbone models. What languages are included in X->E and E->X in Table g1-1 and Table g1-2? Did the authors experiment with low-resource languages like Uzbek, or relatively less challenging low-resource languages like Icelandic?
> > >
> > > Regarding inference speed, I find it intriguing. How did the authors implement "stopping generation at a delimiter token"? I assume the authors are using Hugging Face's model generation functionality. How did you modify the model.generate() function to stop at specific points? Is this feature already supported by HF? Just curious :) I thought it won't stop until the special symbol like  "the end of the setence"
> > >
> > > Finally, I believe the authors should avoid drawing a strict distinction between continued fine-tuning and multi-task learning in this paper. While they are two methods, they are not entirely separate paradigms and are closely related. The authors restrict themselves to a highly specific scenario where only the target data is accessible, but this is not the practical case. In the real world, you always have the translation data and reasoning data.

---

> ### Author Response · Authors · 2024-11-20
> **Further response to Reviewer pUWZ**
>
> Thank you for your response. I sincerely appreciate the opportunity to engage in a more in-depth discussion with you.
>
> > The 'Awkward Position'
>
> As we stated in Lines 042–043, our goal is to develop models that _seamlessly integrate **strong** translation performance with broader general-purpose utility_. And we successfully achieved it by enhancing the translation performance of instruction-tuned LLMs while preserving most of their general abilities. It is important to clarify that "strong" does not imply "state-of-the-art" (SOTA), and at no point have we claimed achieving SOTA performance as either a goal or a contribution of our work. (Frankly, we are surprised that we need to debate about what we wrote in the introduction and the interpretation of this specific word.)
>
> In response to your initial review regrading the gap in translation performance with SOTA LLM-based MT models such as ALMA, we conducted an additional experiment, as presented in General Response 1. The results demonstrate that RaDis, using Qwen-2.5 as its backbone, achieves strong translation performance that surpasses ALMA on the WMT23 test set. Simultaneously, RaDis significantly outperforms ALMA and TowerInstruct in terms of general-purpose capabilities. A 27.91 win rate on Alpaca Eval 2 leaderboard [1] means our model is better than Gemini Pro and Claude 2.1, which are strong chat LLMs. Based on these findings, we believe our claims are well-supported: a model with translation performance surpassing ALMA and general ability outperforming Gemini Pro is not awkward; in fact, it is quite impressive.
>
> Considering the stated goal of our paper, we believe a more appropriate question to evaluate our work would be: If an end user seeks a versatile model that is good at translation, provides strong chat capabilities, and can assist with a wide range of other potential tasks, would they choose ALMA, Llama-2-Chat, or RaDis? We feel this question better aligns with the intended scope and contributions of our work. In practice, such a model could indeed achieve better interactivaty. With strong instruction-following capabilities, RaDis can adapt translations to a specified style, incorporate specific terms as requested, or even provide explanations for translated sentences. These features are beyond the capabilities of traditional LLM-based MT models.
>
> While it is always easy to request SOTA performance on individual tasks, building a model that achieves SOTA across every task simultaneously is highly non-trivial. Although RaDis does not achieve SOTA performance in every task, it successfully enhances the translation performance of instruction-tuned LLMs, creating a model that is "strong" across various tasks—a balance that aligns with the overarching goal of our work and meets the need for general-purpose LLM.
>
> P.S. After reading the response from the reviewer, we still find the phrase "awkward position" to be unclear. In this response, we have tried to address this concern by elaborating on both the performance and practical advantages of our approach. We kindly request that the reviewer provide a more specific criterion or clarify what defines an "awkward" versus "not awkward" position. This would greatly help us better understand and address your concerns.
>
> > Translation performance on low resource languages.
>
> First, we would like to clarify that the experiment in Table g1-1 was conducted under exactly the same settings described in our paper, using Czech, German, Russian, and Chinese. For Table g1-2, we followed the experimental setup outlined in the TowerInstruct paper and directly referenced the results reported there. The evaluation in Table g1-2 was performed on German, Russian, and Chinese, as TowerInstruct was trained on the following 10 languages: English (en), German (de), French (fr), Dutch (nl), Spanish (es), Portuguese (pt), Russian (ru), Chinese (zh), Korean (ko).
>
> Given that RaDis is a post-training method, its performance is inherently influenced by the language proficiency of the backbone LLM. As a result, achieving improved translation performance on low-resource languages not included in the backbone LLM's training set is particularly challenging.
>
> However, we want to emphasize that the goal of our experiment with Qwen-2.5 was to demonstrate that, when equipped with a stronger multilingual backbone, the performance gap caused by additional multilingual pre-training can indeed be reduced. This is an important finding because multilingual backbone LLMs are continually evolving and becoming more capable. As such, we believe our method holds strong potential for future advancements.
>
> > Inference speed.
>
> We conducted inference experiments using vLLM [2]. This framework includes a 'stop/stop_token_ids' parameter, which allows the generation process to halt when specific strings or tokens are produced.
>
> [1] https://tatsu-lab.github.io/alpaca_eval/
>
> [2] https://docs.vllm.ai/en/latest/dev/sampling_params.html

---

> ### Author Response · Authors · 2024-11-20
> **Further response to Reviewer pUWZ (Cont.)**
>
> > Learning paradigms and the accessibility of training data.
>
> First, it is generally acknowledged that continual learning is recognized as a distinct learning paradigm [3,4]. We used this term to align with established terminology in prior works within this field.
>
> Second, while the reviewer claims that task-specific data is always accessible, this is not the case. As we stated in Lines 047-049, the training data for open-sourced chat LLMs is frequently unavailable, making multi-task training with the original data infeasible. This limitation was a primary motivation for us to explore the continual learning paradigm.
>
> Besides, our experiments with open-sourced instruction data clearly demonstrate that relying solely on these datasets is insufficient to effectively mitigate the forgetting of general capabilities. In contrast, our approach using continual learning largely preserves the general capabilities of chat LLMs and outperforms the multi-task training baseline requested, which highlights a significant advantage of our method.
>
> Finally, we believe it is important to note that, although our evaluation focused on instruction following, safety, and math reasoning, general ability encompasses a wide range of tasks (e.g., code generation, semantic classification, summarization, etc.). While it is possible to incorporate additional task-specific datasets into training to improve performance on individual tasks, this approach comes with significant costs, including the effort required to collect such datasets and the computational expense of fine-tuning to achieve balanced performance across tasks.
>
> In contrast, our method avoids these unnecessary training costs by leveraging continual learning to preserve the knowledge embedded in backbone LLMs. This aligns with one of the key motivations behind the continual learning paradigm: to retain knowledge in pre-trained models while integrating new information.
>
> From a broader perspective, RaDis has the potential to serve as a general-purpose continual instruction tuning method, a point that was notably recognized by Reviewers JdnZ and M6wr.
>
> We hope this clarification highlights the strengths and potential of our approach and may encourage you to reconsider our contribution in a more positive light. Thank you again for your valuable insights and constructive feedback—they have been instrumental in helping us better articulate our work.
>
> [3] Liu, Bing. "Lifelong machine learning: a paradigm for continuous learning." Frontiers of Computer Science 11.3 (2017): 359-361.
>
> [4] Wang, Liyuan, et al. "A comprehensive survey of continual learning: theory, method and application." IEEE Transactions on Pattern Analysis and Machine Intelligence (2024).

---

> > ### Comment · Reviewer_pUWZ · 2024-11-20
> >
> > Thank the authors for their quick response :)
> >
> > The term "awkward position" here literally means an "embarrassing position." It's like the authors aim to improve both A and B simultaneously, but in doing so, the method ends up being weaker than models specifically optimized for either A or B. After reviewing the rebuttal, It makes more sense to me that the authors are not trying to create a Sota but rather to find a good trade-off, striving for good performance on both sides without making one aspect significantly worse.
> >
> > The paper still has some issues. As the authors noted, the method heavily depends on the backbone model, which could result in inconsistent and unstable improvements when scaling model size. While the goal of the paper is good, the engineering-focused approach does not deliver particularly impressive results and lacks sufficient novelty to meet the criteria for a higher score (I hate to say it, but it is true in this case).
> >
> > Finally, I acknowledge the findings presented in the paper and the effort reflected in the authors' response. Based on this, I am raising my score to 5.

---

> ### Author Response · Authors · 2024-11-21
>
> Thank you for getting back to me so quickly. We are glad to see that you have raised the score. We would like to take this opportunity to provide additional information to address your remaining concerns.
>
> > Dependency on backbone LLM.
>
> There still seems to be some misunderstanding. As we stated in our previous response, RaDis is a post-training method, and it is generally acknowledged that the language proficiency of the pre-trained LLM influences post-training methods. In this context, dependency on the backbone LLM primarily affects its language proficiency.
>
> > Robustness against scaling model size.
>
> Regarding the new concern raised by the reviewer—robustness against scaling model size—we refer to `Table g1-2` in General Response 1, where we have demonstrated that switching to a stronger backbone (Qwen2.5) leads to improvements across all tested tasks. Additionally, we have implemented RaDis on three different LLMs—Llama-2-Chat, Mistral-v0.2-Instruct, and Qwen2.5-Instruct—and across two different tasks (machine translation and code generation). The performance of RaDis remains highly consistent in these settings. These results clearly indicate that RaDis is a robust approach that generalizes well to different backbone models and tasks. Based on this evidence, we expect similar robustness against scaling model size.
>
> We are happy to provide additional experimental results on scaling model size if the reviewer insists.
>
> > Whether the performance is impressive.
>
> As we have reached an understanding that achieving SOTA performance is not a requirement for this work, we would like to highlight the impressive performance of our method from the perspective of LLM-based MT models, which appears to be a major focus of the reviewer.
>
> As shown in Table g1-2, RaDis consistently outperforms TowerInstruct—a SOTA LLM-based MT model with general instruction-following capabilities—on general tasks, regardless of the backbone utilized. Specifically, RaDis demonstrates a +34% win rate on AlpacaEval, +69% safety improvement on AdvBench, and +81% accuracy improvement on GSM8K. While achieving this superior performance on general tasks, RaDis also delivers strong translation performance, outperforming ALMA. Additionally, all these results are achieved with only 1% of the training cost compared to TowerInstruct.
>
> Taken together, RaDis provides an impressive LLM-based MT model that delivers SOTA performance on general instructions (when focusing on LLM-based MT models specifically) while maintaining strong translation capabilities, all while utilizing over 100x less training compute.
>
> > The novelty and contribution of our work.
>
> We respectfully disagree with the reviewer's claim that our approach is engineering-focused and would like to emphasize that our paper made several unique contributions:
>
> _First_, as demonstrated in General Response 1, RaDis introduces a **novel approach for building LLMs that combine translation proficiency with general abilities**. Previous works (e.g., BayLing, TowerInstruct) primarily rely on a **multi-task training paradigm**, fine-tuning a base LLM on both translation and general instruction data simultaneously. In contrast, RaDis adopts a **continual learning (CL) paradigm**, fine-tuning chat LLMs on translation data while employing continual learning techniques to mitigate the forgetting of general abilities. Our experiments highlight the superiority of this approach in terms of broad general abilities and training efficiency, while also showcasing its competitive performance and promising potential in translation tasks.
>
> _Second_, RaDis introduces a **novel Continual Instruction Tuning (CIT) method with a strong motivation**. To the best of our knowledge, we are the first to observe that LLMs tend to generate detailed post-rationales even without explicit chain-of-thought (CoT) prompting. Furthermore, we are the first to leverage this observation by proposing a CIT method that distills self-generated rationales. Our experiments demonstrate that RaDis is a robust CIT approach, capable of generalizing effectively across different backbone models and tasks. We believe introducing rationales to CL brings new insights into relying on the **reasoning ability** of LLMs to **connect learned and new knowledge** and alleviate forgetting.
>
> Given these points, we believe your concerns regarding the novelty of our work can be effectively addressed. Nonetheless, we will further emphasize our contributions in the revised version of the paper. If there are specific concerns regarding our contributions—such as previous works that may have addressed similar ideas—we kindly ask the reviewer to provide those references so that we can better understand and address these concerns.

---

> ### Author Response · Authors · 2024-12-03
> **Kind reminder**
>
> Dear Reviewer pUWZ,
>
> Thank you once again for the time and effort you've dedicated to reviewing our work. As the discussion phase comes to a close, we kindly ask that you review our follow-up responses (including the new general responses), which may address the questions raised.
>
> **While we acknowledge any differences in opinion, we deeply appreciate the thoughtful feedback you’ve provided. We understand that multiple rounds of detailed review are rare and invaluable, and our goal with these responses is to foster a more thorough discussion of differing viewpoints. Regardless of whether you choose to maintain or revise your score, we fully respect your decision.**
>
> Best regards,
>
> Authors

---

### Author Response · Authors · 2024-11-19
**General Response 1: On Request for Additional Comparisons to TOWER and Fine-tuning with Instruction Data**

We want to thank the reviewers for their thoughtful comments. We are diligently working to provide comprehensive responses that address each concern, and we will continue to present concrete experimental results throughout the discussion period.

In the following, we address specific points related to the baselines the reviewers have requested comparisons to—namely, multi-task training with additional instruction data and the Tower models.

First, we would like to clarify that RaDis is a **Continual Instruction Tuning (CIT)** method. In a **Continual Learning (CL)** setting, learning is done incrementally. When learning a new task, only the data from the task is available, and the model is trained to adapt to this task without access to the data of previous tasks. On the other hand, fine-tuning with additional instruction data corresponds to a **multi-task learning** approach, which learns multiple tasks simultaneously or jointly. It is regarded as the upper bound for CL methods. Therefore, the requested comparisons are actually between **two distinct paradigms**: a multi-task learning one and a continual learning one. Besides, we would also like to emphasize that both ParroT and BayLing have included instruction data (Alpaca) in model training. Therefore, models built in the multi-task training paradigm are already included in our original experiments.

However, we are glad to report that our RaDis outperforms the multi-task fine-tuning baseline in terms of preserving the general abilities of the LLMs, and performs competitively in translation.

Table g1-1: Comparison to multitask training. The best result in each column is marked in **bold**.
|                       | X-EN      | EN-X      | Alpaca Eval | Alpaca Eval 2 | AdvBench  | GSM8K     |
|-----------------------|-----------|-----------|-------------|---------------|-----------|-----------|
| Llama-2-Chat          | 79.01     | 73.39     | 71.40       | 9.66          | 100       | 21.83     |
| +Multitask            | **81.75** | 82.59     | 44.55       | 3.98          | 98.65     | 11.98     |
| +RaDis                | 81.22     | **82.61** | **67.94**   | **7.47**      | **100**   | **19.48** |
| Mistral-v0.2-Instruct | 80.84     | 67.79     | 84.91       | 15.09         | 68.46     | 41.62     |
| +Multitask            | **82.09**    | **84.44** | 49.46       | 5.45          | **63.85** | 22.97     |
| +RaDis                | 81.94 | 84.39     | **80.34**   | **11.05**     | 62.12     | **41.70** |

`Table g1-1` presents a comparison with the multi-task training baseline (+Multitask). Specifically, we utilize two widely adopted instruction datasets, Alpaca [1] and Dolly [2], and add them into fine-tuning. As shown in the table, our RaDis consistently outperforms the multi-task training baseline in preserving the general abilities of the LLM, while achieving comparable performance on translation tasks.

This result aligns with our claim in Lines 109-113: the quality of open-sourced instruction data is generally lower than the in-house data used for training large language models like Llama or Mistral. Consequently, incorporating these open-sourced datasets into training does not effectively mitigate the forgetting of the LLM's general abilities. Moreover, since the open-sourced data is out-of-distribution relative to the backbone LLM, fine-tuning with these data may even exacerbate the forgetting problem [3].


[1] Stanford Alpaca: An Instruction-following LLaMA model. https://github.com/tatsu-lab/stanford_alpaca

[2] Conover, Mike, et al. "Free dolly: Introducing the world’s first truly open instruction-tuned llm." Company Blog of Databricks (2023).

[3] Yang, Zhaorui, et al. "Self-distillation bridges distribution gap in language model fine-tuning." Proceedings of the 62nd Annual Meeting of the Association for Computational Linguistics (Volume 1: Long Papers). (2024).

---

> ### Author Response · Authors · 2024-11-19
> **On Request for Additional Comparisons to TOWER and Fine-tuning with Instruction Data (Cont)**
>
> Table g1-2: Comparison to TowerInstruct-v0.2. The best result in each column is marked in **bold**. The second best is _italicized_.
>
> |                    | X-EN      | EN-X      | Alpaca Eval | Alpaca Eval 2 | AdvBench  | GSM8K     |
> |--------------------|-----------|-----------|-------------|---------------|-----------|-----------|
> | Mistral+RaDis      | 80.64     | 80.58     | _80.34_     | _11.05_       | _62.12_   | _41.70_    |
> | Qwen2.5+RaDis      | _82.13_   | _82.81_   | **85.62**   | **27.91**     | **99.04** | **88.78** |
> | ALMA | 81.65 | 81.91 | 1.08       | 0.17          | -    | 0  |
> | TowerInstruct-v0.2 | **82.77** | **84.28** | 51.59       | 4.02          | 30.19     | 7.35      |
>
> Given that TowerInstruct has been fine-tuned on the WMT22 test set, we shifted the translation test set to the WMT23 test set. We report the performance of the best model in our paper (Mistral+RaDis) alongside TowerInstruct. To further demonstrate the potential of our approach, we also conducted new experiments with Qwen2.5-Instruct as the backbone (Qwen2.5+RaDis).
>
> As shown in `Table g1-2`, RaDis consistently outperforms TowerInstruct-v0.2 in terms of preserving general abilities. This is primarily because TowerInstruct-v0.2 is fine-tuned using UltraChat, which, like other open-sourced instruction datasets, suffers from lower quality.
>
> In terms of translation, TowerInstruct-v0.2 achieves higher performance, largely due to the benefits of multilingual pre-training and extensive parallel fine-tuning. However, we would like to emphasize the strong potential of our approach from two key perspectives:
>
> 1. **RaDis is more efficient**: The training times for TowerBase 7B and 13B were 80 and 160 GPU days, respectively, using A100-80GB GPUs. Fine-tuning TowerInstruct adds an additional 200 GPU hours. In contrast, RaDis requires only 20 GPU hours (4 hours for generating rationales and 16 hours for training), which is **less than 1%** of the training cost for TowerInstruct-7B, while still achieving strong performance.
> 2. **RaDis can benefit from stronger backbone LLM**: While TowerInstruct achieves better translation performance, RaDis can effectively bridge this gap by leveraging a stronger backbone LLM. As shown in `Table g1-2`, switching the backbone from Mistral to Qwen2.5 leads to substantial improvements across all tasks and outperforms ALMA. We believe that as open-source multilingual LLMs continue to improve, the performance gap in translation will gradually narrow.
>
> Together, these results underscore the advantages of our approach and demonstrate that RaDis offers a novel and competitive paradigm for building LLMs that excel in both translation proficiency and general ability.
>
> Following the reviewers' suggestion, we will include the additional experiments and discussions comparing RaDis with TowerInstruct and the multi-task training baseline in the revised version of our paper.

---

### Author Response · Authors · 2024-11-19
**General Response 2: On Request for Additional Experiments on Other Tasks**

We appreciate Reviewer JdnZ and M6wr for recognizing the potential of RaDis, and we would be happy to explore its application across a broader range of tasks. Due to time constraints, we have currently only completed experiments on the code generation task. Specifically, we fine-tuned Mistral-v0.2 on Python code data from the Magicoder dataset and evaluated its performance using HumanEval and general ability benchmarks.

Table g2: Experiments on code generation. Pass@1 is reported on HumanEval.
|              | HumanEval | Alpaca Eval | Alpaca Eval 2 | AdvBench | GSM8K |
|--------------|-----------|-------------|---------------|----------|-------|
| Mistral-v0.2 | 36.59     | 84.91       | 15.09         | 68.46    | 41.62 |
| +Vanilla-FT  | 42.07     | 73.89       | 8.75          | 40       | 43.97 |
| +SDFT        | 41.24     | 78.58       | 10.46         | 48.08    | 40.86 |
| +RaDis       | 43.9      | 80.25       | 11.4          | 51.92    | 42.91 |

As shown in `Table g2`, RaDis outperforms Vanilla-FT and SDFT in code generation tasks, achieving higher Pass@1 on HumanEval and excelling in other benchmarks for general abilities.

A key reason for this is that RaDis directly preserves the original references in the dataset, whereas SDFT paraphrases them. Intuitively, while paraphrasing helps bridge the distribution gap, it also reduces the amount of learnable knowledge. As a result, SDFT may struggle to outperform Vanilla-FT on certain tasks. In contrast, RaDis directly utilizes the original references, preserving all the knowledge embedded in the data.

Regarding performance on general tasks, RaDis still outperforms SDFT. We believe this can be attributed to the distribution gap. While SDFT claims to distill the dataset, it paraphrases the data. As a result, the model's responses are sampled from the paraphrased instruction's output distribution, which tends to be out-of-distribution relative to the original task instruction. In contrast, RaDis performs self-distillation using rationales, which are fully in-distribution. This enables RaDis to more effectively alleviate forgetting and better preserve general abilities.

These results suggest that RaDis generalizes well to a broader range of tasks, highlighting its potential as a robust, general-purpose continual instruction tuning method. We plan to investigate this potential in future works.

---

> ### Author Response · Authors · 2024-12-03
> **General Response 2: On Request for Additional Experiments on Other Tasks (Cont.)**
>
> We have conducted additional experiments on TriviaQA (open-ended QA) and OpenFunctions (function call). Below are the results from these experiments:
>
> |                 | **OpenFunctions** | **Alpaca Eval** | **Alpaca Eval 2** | **AdvBench** | **gsm8k** |
> |-----------------|-------------------|-----------------|-------------------|--------------|-----------|
> | **Llama-2**     | 16.07             | 71.4            | 9.66              | 100          | 24.34     |
> | **+Vanilla-FT** | 29.46             | 36.52           | 3.41              | 100          | 20.69     |
> | **+SDFT**       | 32.14             | 68.51           | 7.82              | 100          | 21.00        |
> | **+RaDis**      | **33.04**         | **69.52**       | **8.79**          | 100          | **24.56** |
>
> |                  | **TriviaQA** | **Alpaca Eval** | **Alpaca Eval 2** | **AdvBench** | **gsm8k** |
> |------------------|--------------|-----------------|-------------------|--------------|-----------|
> | **Mistral-v0.2** | 53.4         | 84.91           | 15.09             | 68.46        | 41.62     |
> | **+Vanilla-FT**  | 57.9         | 29.82           | 3.09              | 30.38        | 9.86      |
> | **+SDFT**        | 53.83        | 79.47           | 10.4              | 36.35        | **40.64** |
> | **+RaDis**       | **58.71**    | **83.95**       | **14.22**         | **64.81**    | 39.27     |
>
> As demonstrated in the tables above, RaDis effectively alleviates forgetting and outperforms SDFT on both tasks. Additionally, it achieves superior performance on the fine-tuned task compared to Vanilla-FT. These results support our claim that rationales play a crucial role in bridging learned and new knowledge, thereby enhancing both task learning and knowledge retention.

---

### Author Response · Authors · 2024-11-25
**General Response to AC and Reviewers**

Dear AC and reviewers,

Thank you for your insightful suggestions. We truly appreciate the time and effort you’ve invested in reviewing our work. In response to your feedback, we have refined our experiments and incorporated additional analyses. **The revised version of the paper has been uploaded**, and we summarize the key modifications below:

1. We have introduced a **multi-task training** baseline, allowing for a more comprehensive comparison with this straightforward approach. (See General Response 1)
2. We discuss and compare our work with **TowerInstruct**, the state-of-the-art LLM-based MT model with general capabilities. (See General Response 1)
3. We conducted experiments on broader tasks to assess the **generalizability** of our proposed method. (See General Response 2)
4. We have corrected minor typos and clarified certain claims to improve overall clarity. (See General Response)

As the discussion phase is going to approach its conclusion, we would appreciate any feedback on whether our revisions address your concerns. We look forward to hearing from you.

Best regards,
The Authors

---

### Author Response · Authors · 2024-11-29
**General Response 3: Continual Learning (CL) v.s. Multi-task Learning (MTL)**

We hope to provide more information regarding the differences between CL and MTL, and the practical needs for CL in the era of LLMs.

First, we would like to clearify that **continual learning** is different from **continual fine-tuning**. CL refers to the broader concept of learning over time, where a model is designed to learn from a stream of data in an ongoing fashion, without forgetting previously acquired knowledge. It  is recognized as a novel learning paradigm [1,2,3]. In contrast, continual fine-tuning represents fine-tuning a model with new task data, it does not involve strategies to alleviate forgetting and is considered a baseline approach in CL (Vanilla-FT).

> Differences between CL and MTL

While both CL and MTL aim to improve a model's ability to generalize, CL is focused on learning over time without forgetting past knowledge, whereas MTL focuses on learning multiple tasks concurrently.

CL is essential for intelligence, as humans learn continuously in an ongoing manner. For instance, today we might encounter pigs and apples and learn to recognize them. Tomorrow, we might see sheep and learn to identify them as well. This incremental learning process is not suitable for MTL because when we first see pigs and apples, the sheep have not yet appeared. It would be inefficient or even impossible to revisit the images of pigs and apples and relearn them alongside sheep when they eventually appear, as MTL would require.

> CL is widely needed in LLM applications.

A similar situation arises in the context of LLM applications. Consumers typically aim to adapt open-sourced LLMs for specific applications (eg. health care, education). Preserving the original knowledge in these LLMs is essential for achieving optimal performance. However, fine-tuning with domain- or task-specific data often leads to significant forgetting. In this situation, MTL is often infeasible, as we lack access to the data used to train the LLM. This real-world challenge is one of the key motivations behind our decision to work on this problem and it highlights the requirements for CL approaches in LLM applications [3].

[1] Liu, Bing. "Lifelong machine learning: a paradigm for continuous learning." Frontiers of Computer Science 11.3 (2017): 359-361.

[2] Wang, Liyuan, et al. "A comprehensive survey of continual learning: theory, method and application." IEEE Transactions on Pattern Analysis and Machine Intelligence (2024).

[3] Shi, Haizhou, et al. "Continual learning of large language models: A comprehensive survey." arXiv preprint arXiv:2404.16789 (2024).

---

### Author Response · Authors · 2024-11-29
**General Response 4: The Novelty of Utilizing Rationales in CL**

We hope to suggest that the use of rationales to alleviate catastrophic forgetting in RaDis represents a **novel approach** in the CL field. Traditional replay-based methods need to retain previous task examples and replay them during fine-tuning. In contrast, RaDis solely leverages **self-generated rationales**—the model's own explanations for new training data. These rationales go beyond raw outputs; they introduce a form of **model introspection**, where the model uses its internal knowledge to explain an external example.

In that case, these rationales serve as a **semantic scaffold** that connects learned knowledge with new tasks, building a reasoning framework that ties internal concepts together, which helps both **generalization** and **knowledge retention**. This mechanism mirrors **how humans often retain information**—not simply by memorizing facts (**replay example**), but by constructing a mental narrative (**rationale**) that explains how new knowledge fits within a broader context. By utilizing rationales, RaDis offers the potential for a more **human-like memory system**, representing a promising new direction for continual learning.

We sincerely hope the clarification above could help the reviewer to focus on the insightful ideas behind RaDis and reconsider their scores.

---

### Meta-Review · Area_Chair_bDMM · 2024-12-23

**Metareview:**

### Summary
The paper introduces RaDis (Rationale Distillation)  to improve machine translation capabilities of large language models (LLMs) without compromising their general instruction-following abilities. The approach leverages self-generated rationales to mitigate catastrophic forgetting while enabling efficient training with minimal computational resources. Extensive experiments demonstrate competitive performance in translation while preserving general task abilities, highlighting its potential for versatile LLM applications.

### Strengths
The motivation of using self-generated rationales to balance specialized and general capabilities is good. Extensive empirical results are shown across multiple models and tasks, showcasing improved efficiency over traditional methods.

### Weaknesses:
Main weakness is that the paper is a bit limited in its scope and the proposed method itself doesn't bring new insights to the post-training research community of LLMs.

**Additional Comments On Reviewer Discussion:**

During the rebuttal period, the following key points are raised:

1. Applicability across tasks: Reviewers recommended broader experiments beyond translation, such as QA or summarization. Authors responded with additional code generation experiments but acknowledged time constraints for further tests.

2. Inference Efficiency: Concerns about output generation overhead were raised. The authors explained techniques like stopping generation at a delimiter token to mitigate issues, showing efficiency improvements.

3. Comparison Baselines: Reviewers pointed out unfair baselines and missing baselines like training with mixed datasets. The authors provided updated comparisons, demonstrating competitive performance relative to multi-task learning setups.

---

### Decision · Program_Chairs · 2025-01-22

Reject